# Simplifying Actor-Critic Reinforcement Learning: Mitigating Overestimation Bias with a Single Distributional Critic

## Abstract

Actor-critic methods in reinforcement learning leverage the action value function (critic) by temporal difference learning to be used as an objective for policy improvement for the sake of sample efficiency against on-policy methods. The well-known result, critic overestimation, is usually handled by pessimistic policy evaluation based on critic uncertainty, which may lead to critic underestimation. This means that pessimism is a sensitive parameter and requires careful tuning. Current methods use epistemic or predictive uncertainty of the critic for pessimistic learning, employing dropout and ensemble approaches. In this paper, we propose a novel actor-critic algorithm, called Stochastic Actor-Critic (STAC), that employs distributional representation (for aleatoric uncertainty) and Bayesian dropout (for epistemic uncertainty) for critic and actor to make the agent uncertainty aware. Unlike previous methods, pessimistic updates are only proportional to aleatoric uncertainty of the critic, not epistemic uncertainty. This approach alone is enough to mitigate critic overestimation. Introducing Bayesian dropout further improves performance in some environments, although the resulting uncertainty is not used for pessimistic objective. With empirically determined optimal pessimism and dropout rate, only a single distributional critic network is enough to achieve high sample efficiency. In addition, using a single critic with an update-to-data (UTD) ratio equal to 1 provides computation-efficient learning compared to other SOTA methods.

## 1 Introduction

Reinforcement learning (RL) has witnessed significant progress with the emergence of deep neural networks (Mnih et al., 2013; 2015) in the last decade. However, sample efficiency is one of the main bottlenecks of widespread adaptation of RL into applications (Mendonca et al., 2019; Li et al., 2023a). On the other hand, computation efficiency is as important as sample efficiency (Chen et al., 2021a), especially when RL agents are deployed in real-life applications such as robots (Zhao et al., 2020; Kormushev et al., 2013) and edge devices (Dai et al., 2022; Wei et al., 2022).

Actor-critic methods leverage off-policy samples to train critics, promising higher sample-efficient learning than on-policy algorithms. Despite this advantage, they are usually stuck on poor performance due to a mismatch between behavioral and online policy. Critic is trained by samples from behavioral policy, while it is expected to evaluate on-policy actions. Therefore, critic estimates include errors naturally, and these erroneous estimates are exploited by policy, and the algorithm suffers from critic overestimation that produces divergent (catastrophic) behavior (Thrun & Schwartz, 2014). This problem is also named as *deadly triad* ((Sutton & Barto, 2018; Van Hasselt et al., 2018)) indicating the instability emerges once function approximation, temporal difference, and off-policy learning are in the same method together. In the literature, main solution to the critic overestimation problem is to use a pessimistic learning objective. Such models use lower bounds (conservative estimates) as the objective at the cost of underexploration.

## 1.1 Related Work

**Pessimism upon epistemic uncertainty**  In the literature, the overestimation problem is mainly solved by pessimistic learning based on *epistemic uncertainty* of critic (Chen et al., 2021b; Hiraoka et al., 2021). The same approach is also used in model-based RL methods (Janner et al., 2019; Chua et al., 2018; Depeweg et al., 2016). Recently, for off-policy model-free actor-critic algorithms, epistemic uncertainty is estimated by using double network (Fujimoto et al., 2018; Haarnoja et al., 2018) or ensemble network (Chen et al., 2021b; Moskovitz et al., 2021). Ensemble methods are computationally expensive as there are multiple of parameters to be optimized. For this, Hiraoka et al. (2021) found out that using Bayesian dropout (Srivastava et al., 2014; Gal & Ghahramani, 2016) also contributes to epistemic uncertainty assessment but a small ensemble is still required. On the contrary, He et al. (2021) argued that a single critic network is enough when dropout is also used to evaluate Bellman backup. The mentioned methods use a constant level of pessimism for policy evaluation and improvement. Moskovitz et al. (2021) focused on updating pessimism *on the fly* as a bandit problem rather than fixing it. They also argued that optimal pessimism (or optimism) depends on the environment, task, and learning method. Li et al. (2023b) goes beyond this approach by parameterization of optimism/pessimism with a neural network and obtains significantly good performance on benchmark environments.

**Pessimism upon aleatoric uncertainty**  Aleatoric uncertainty representation for the value function carries fundamental importance, especially in the presence of approximation (Bellemare et al., 2017). This can be conducted by atoms (Bellemare et al., 2017), quantiles (Dabney et al., 2018), and a probability distribution (Tang et al., 2019; Yang et al., 2021). Kuznetsov et al. (2020) claims *aleatoric uncertainty* is also responsible for overestimation since any randomness is exploited when the Bellman optimality operator ($\mathcal{T}^*$) is employed. For this purpose, they use ensemble networks for epistemic uncertainty, in which each network is a distributional network (as quantiles) for aleatoric uncertainty assessment. Their algorithm uses both types of uncertainties for overestimation correction. For safety-critical RL applications to avoid catastrophic situations, pessimistic policy updates upon aleatoric uncertainty are modeled as normal distribution by Tang et al. (2019) and Yang et al. (2021). Stachowicz & Levine (2024) devised a risk-sensitive actor-critic algorithm in which epistemic uncertainty is modeled by ensemble whereas aleatoric uncertainty is modeled by distributional representation as an output of critic network similar to the work of Kuznetsov et al. (2020). Their approach leads to higher performance by significantly reducing unsafe maneuvers.

**Dropout uncertainty**  Using dropout is a kind of Bayesian approximation, so another way to assess epistemic uncertainty (Gal & Ghahramani, 2016). It has applications on model-based (Gal et al., 2016a; 2017; Kahn et al., 2017) and model-free (Moerland et al., 2017; Jaques et al., 2019; He et al., 2021) reinforcement learning. Using the same idea in off-policy maximum entropy actor-critic setting, He et al. (2021) injects dropout to the critic network and demonstrates that one critic network is enough for an actor-critic method. Similarly, Hiraoka et al. (2021) uses the dropout mechanism to evaluate epistemic uncertainty in addition to ensembling and shows that it reduces the number of required networks in the ensemble but they used critics with deterministic outputs. Dropout allows for significantly reduced ensemble size and still uses a high UTD ratio. They also experimented with a single critic network (called Sin-DroQ) and obtained similar performance only in easy environments but failed to converge in difficult ones.

**Optimism in the face of uncertainty**  Epistemic uncertainty is also employed to improve policy in *optimistic* manner (Audibert et al., 2007; Kocsis & Szepesvári, 2006). This principle provides a reasonable exploration scheme in an on-policy setting as this encourages exploration of state-action space. For large-scale problems, this approach either fails or requires carefully tuned optimism (Pacchiano et al., 2020; Ciosek et al., 2019). O'Donoghue et al. (2018) used normal distribution to track critic uncertainty in which the upper bound is used as a policy improvement target. Osband et al. (2016) follows a similar way but uses ensembles, and improves policy with random critics at each episode inspired by Thompson sampling. In the actor-critic setting, Tasdighi et al. (2024) and Ciosek et al. (2019) implemented a double critic network and used optimistic estimates for policy improvement while constructing pessimistic critic targets for policy evaluation to mitigate the critic overestimation problem. Unlike others, Gal et al. (2016a; 2017); Azizzadenesheli et al. (2018); Wu et al. (2023) employed Bayesian dropout for optimistic exploration.

## 1.2   Stochastic Actor-Critic Algorithm

In this paper, we introduce a novel off-policy actor-critic algorithm, Stochastic Actor-Critic (STAC), specifically designed to address both sample and computation inefficiencies. STAC is an off-policy maximum entropy actor-critic algorithm employing experience buffer to sample off-policy transition tuples, similar to Soft Actor-Critic (SAC) (Haarnoja et al., 2018).

As the first contribution, we discuss whether using only aleatoric uncertainty is enough to be used for pessimistic learning, instead of epistemic uncertainty. Although it is defined as the inherent stochasticity of the process, it also arises due to the agent's lack of learning capability. Therefore, there should be aleatoric uncertainty even within deterministic environments. For this purpose, we define critic as a distributional (heteroscedastic) model, which also allows learning loss attenuation, making the critic loss more robust to noisy data (Kendall & Gal, 2017).

Our second contribution is a theoretical analysis of critic overestimation under the maximum entropy actor-critic framework. As a result of this analysis, STAC devises environment-specific pessimism hyper-parameter to be used upon *aleatoric uncertainty* of critic. Pessimistic updates are conducted in both policy evaluation and improvement phases to tackle critic overestimation. The optimal pessimism improves sample efficiency, yielding similar results to other methods that use a much higher update-to-data (UTD) ratio and ensembles.

Epistemic uncertainty can be used for exploration, which is the main idea of optimism in the face of uncertainty principle. Most actor-critic methods use epistemic uncertainty for pessimism to mitigate critic overestimation, revealing the necessity of a high update-to-data (UTD) ratio. This is also the main reason for the difficulty of devising an optimistic actor-critic algorithm in our opinion. STAC allows not to use *epistemic uncertainty* for overestimation mitigation, which would hinder exploration of state-action space of the problem. Instead, it employs *pessimism in the face of aleatoric uncertainty* for this problem. STAC employs *Bayesian dropout* (Srivastava et al., 2014; Gal & Ghahramani, 2016), which is also used for epistemic uncertainty in the literature. However, dropout is not used for pessimistic learning but for exploration and regularization. Using dropout at the policy improvement phase inherently conducts Thompson sampling (Gal & Ghahramani, 2016; Gal et al., 2016a; 2017) in a heuristic way.

The implementation is very simple and can be obtained by injecting dropout to networks, introducing a distributional critic network, and defining and pessimistic learning objective upon the well-known Soft Actor-Critic algorithm (Haarnoja et al., 2018), without a double critic network. We conduct extensive experiments on standard RL benchmarks to evaluate the performance of STAC compared to existing methods. Our results demonstrate the effectiveness of STAC in achieving competitive performance to SOTA methods while requiring fewer computational resources and fewer samples, making it a promising approach for real-world RL applications.

## 2   Reinforcement Learning Preliminaries

This section is dedicated to briefly explain the reinforcement learning concept and actor-critic methodology. Throughout the paper, $\mathcal{P}(\Omega)$ denotes the set of all possible probability distributions on set $\Omega$.

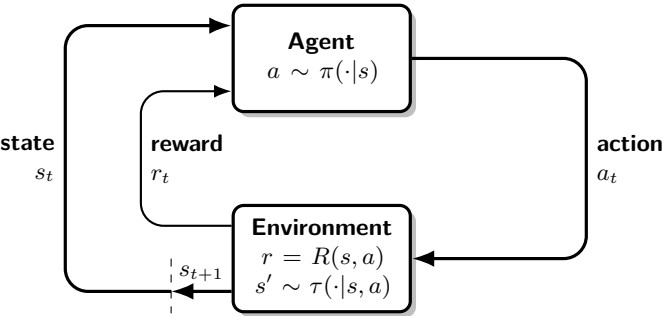

Figure 1: Markov Decision Process (MDP) Loop.

## 2.1 Model-free Reinforcement Learning

In reinforcement learning language, the agent lives in a Markov Decision Process (MDP) which is represented by a tuple $\mathcal{M} = (\mathcal{S}, \mathcal{A}, d_0, \tau, R)$, where $\mathcal{S}$ is state space, $\mathcal{A}$ is action space, $d_0 \in \mathcal{P}(\mathcal{S})$ is initial state distribution, $\tau : \mathcal{S} \times \mathcal{A} \to \mathcal{P}(\mathcal{S})$ is transition kernel and $R : \mathcal{S} \times \mathcal{A} \to \mathbb{R}$ is reward function.

The MDP loop is illustrated in Figure 1. The initial state is sampled first, $s_0 \sim d_0(\cdot)$. At each time $t$ being on $s_t \in \mathcal{S}$; next state is obtained from the environment, $s_{t+1} \sim \tau(\cdot \mid s_t, a_t)$ depending on the taken action $a_t \sim \pi(\cdot \mid s_t)$. Finally, a reward is obtained, $r_t = R(s_t, a_t)$ from the reward function $R$. The ultimate goal of the agent is to derive a policy $\pi : \mathcal{S} \to \mathcal{P}(\mathcal{A})$ to maximize discounted cumulative return, i.e., value function for a given state $s$,

$$V^\pi(s) = \mathbb{E}_{\pi,\tau}\Big[ \sum_{t=0}^\infty \gamma^t R(s_t, a_t)\Big|s_0 = s\Big]. \tag{1}$$

## 2.2 Maximum Entropy Actor-Critic

To promote random actions for exploration and algorithm robustness, maximum entropy framework introduces policy entropy bonus into value functions (Haarnoja et al., 2017; 2018),

$$V^\pi(s) = \mathbb{E}_{\pi,\tau}\Big[ \sum_{t=0}^\infty \gamma^t R(s_t, a_t) - \alpha \log \pi(a_t|s_t)\Big|s_0 = s\Big], \tag{2}$$

$$Q^\pi(s,a) = R(s,a) + \mathbb{E}_{\pi,\tau}\Big[ \sum_{t=1}^\infty \gamma^t R(s_t, a_t) - \alpha \log \pi(a_t|s_t)\Big|s_0 = s, a_0 = a\Big]. \tag{3}$$

Learning iterates between solving policy evaluation and policy improvement. For the definition of critic, Bellman backup operator $\mathcal{T}^\pi$ is defined,

$$\mathcal{T}^\pi Q(s,a) = R(s,a) + \gamma \mathbb{E}_{\substack{s' \sim \tau(\cdot|s,a)\\a' \sim \pi(\cdot|s')}}\big[Q(s', a') - \alpha \log \pi(a' \mid s')\big], \tag{4}$$

and the critic is expected to remain same if this operator applied on itself, i.e., $Q^\pi(s,a) = \mathcal{T}^\pi Q^\pi(s,a)$. Policy evaluation minimizes the temporal difference, i.e., the difference between $Q$ and $\mathcal{T}^\pi Q$ to satisfy this condition. In practice, the temporal difference (TD) target is used instead of the Bellman backup $\mathcal{T}^\pi Q(s,a)$ for learning. TD target is calculated by a random action drawn from the policy $\pi$, instead of expectation.

The optimal policy gives the maximum value, i.e., $\pi^*(\cdot|s) = \arg\max_\pi V^\pi(s)$. Policy improvement improves policy by updating estimated critic $Q^k$ at $k$th iteration as follows,

$$\pi^{k+1}(\cdot \mid s) = \arg\min_\pi \mathrm{KL}\Big(\pi(\cdot \mid s)\Big\|\frac{\exp(\alpha^{-1}Q^k(s,\cdot))}{\int_\mathcal{A} \exp(\alpha^{-1}Q^k(s,a))da}\Big). \tag{5}$$

After sufficient iteration, both policy and critic converge to optimality in the ideal case. In this iteration, critic is also updated, $Q^{k+1}(s,a) = \mathcal{T}^* Q^k(s,a)$, where $\mathcal{T}^*$ is the Bellman optimality opertor (Equation 5 from Haarnoja et al. (2017)),

$$\mathcal{T}^* Q(s,a) = R(s,a) + \gamma \mathbb{E}_{s' \sim \tau(\cdot|s,a)}\Big[\alpha \log \Big( \int_\mathcal{A} \exp(\alpha^{-1}Q(s',a'))da'\Big)\Big]. \tag{6}$$

and the optimal critic function satisfies the Bellman optimality condition, $Q^*(s,a) = \mathcal{T}^* Q^*(s,a)$.

## 3 Modeling Aleatoric and Epistemic Uncertainties

In this part, we explain the differences between two main types of uncertainties, *aleatoric* and *epistemic* uncertainty (Der Kiureghian & Ditlevsen, 2009; Kendall & Gal, 2017; Gal et al., 2016b). Most deep learning methods model either epistemic or aleatoric uncertainty alone (Gal et al., 2016b), whereas modeling both has fundamental importance for reliable and robust predictions.

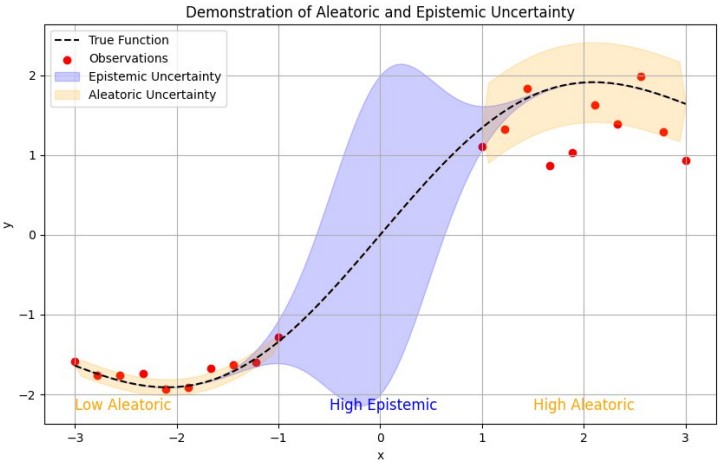

Figure 2: Aleatoric vs Epistemic uncertainty illustration.

## 3.1 Aleatoric Uncertainty

This type of uncertainty comes from the inherent randomness within the data. It is sometimes called statistical or data uncertainty. Even if more data are collected, it is unavoidable and cannot be reduced, because it is an intrinsic part of the process being modeled, including measurement errors or natural variability in the data. In addition, uncertainty due to lack of learning capacity may also appear as aleatoric uncertainty as it cannot be reduced by collecting more data. In other words, the agent cannot decide true deterministic output just because it is not capable of doing it and assigns a non-deterministic distribution as output.

For regression problems in deep learning setting, we can model this by having a distributional (heteroscedastic) network (with parameter $\theta$) that outputs a normal distribution $\mathcal{N}(\mu_\theta(x), \sigma_\theta^2(x))$, where both mean and variance depends on input $x$ (Lakshminarayanan et al., 2017; Kendall & Gal, 2017). Given a dataset $\mathcal{D} = \{(x_i, y_i)\}_{i=1}^N$, the loss function for training the network can be derived from the negative log-likelihood of the normal distribution,

$$\mathcal{L}_\theta(\mathcal{D}) = -\log p(\mathcal{D} \mid \theta) = \frac{1}{N} \sum_{i=1}^N \left( \frac{1}{2\sigma_\theta^2(x_i)} (y - \mu_\theta(x_i))^2 + \frac{1}{2} \log \sigma_\theta^2(x_i) \right) + \frac{1}{2} \log 2\pi. \tag{7}$$

## 3.2 Epistemic Uncertainty

Epistemic uncertainty reflects the uncertainty in the model parameters due to insufficient training data, or incomplete understanding of the underlying process. This kind of uncertainty can be reduced by gathering more data or using a better model with higher generalization capability. The difference between aleatoric and epistemic uncertainty is illustrated in Figure 2. Aleatoric uncertainty is high in inherently random regions while epistemic uncertainty is high where there is no (or less) data.

In deep learning context, Bayesian neural networks (BNNs) provide a way to model epistemic uncertainty. Given the training data $\mathcal{D}$ and prior distribution $p(\theta)$ over the network parameters $\theta$, we can compute the posterior distribution over the parameters $p(\theta \mid \mathcal{D})$ using variational inference.

**Practical Implementation with Monte Carlo Dropout**   Monte Carlo dropout is a practical method to approximate Bayesian inference in neural networks (Gal & Ghahramani, 2016; Gal et al., 2017). Sampled weights $\theta$ are same as network weights $w$ but randomly masked by dropout. During training, dropout is applied and the model learns to make predictions with dropout active. The loss function in this setting typically remains the same as the standard loss (e.g., negative log-likelihood).

## 4 Quantifying Overestimation for Sub-Gaussian Critic Distributions

In this part, we analyze how estimation error causes overestimation due to policy improvement. Assuming the policy improvement step is successful given critic function $Q \in \mathbb{R}^{\mathcal{S} \times \mathcal{A}}$, the target used to update the critic in maximum entropy framework is Bellman backup $\mathcal{T}^{\pi}Q$, i.e., Bellman backup operator applied on $Q$. The definition uses the deterministic function $Q$ (Equation 4) while the critic may only be known with uncertainty, represented as a predictive distribution $\mathcal{Q} \in \mathcal{P}(\mathbb{R}^{\mathcal{S} \times \mathcal{A}})$. Therefore, we define expected Bellman backup $\mathbb{E}_{Q \sim \mathcal{Q}}[\mathcal{T}^{\pi}Q(s,a)]$ as follows;

$$\mathbb{E}_{Q \sim \mathcal{Q}}[\mathcal{T}^{\pi}Q(s,a)] = R(s,a) + \gamma \mathbb{E}_{\substack{Q \sim \mathcal{Q} \\ s' \sim \tau(\cdot|s,a) \\ a' \sim \pi(\cdot|s')}} \Big[ Q(s',a') - \alpha \log \pi(a' \mid s') \Big]. \tag{8}$$

Similarly, we define expected Bellman update $\mathbb{E}_{Q \sim \mathcal{Q}}[\mathcal{T}^*Q(s,a)]$, i.e., Bellman optimality operator applied on $\mathcal{Q}$ as follows;

$$\mathbb{E}_{Q \sim \mathcal{Q}}[\mathcal{T}^*Q(s,a)] = R(s,a) + \gamma \mathbb{E}_{\substack{Q \sim \mathcal{Q} \\ s' \sim \tau(\cdot|s,a)}} \Big[ \alpha \log \Big( \int_{\mathcal{A}} \exp(\alpha^{-1}Q(s',a'))da' \Big) \Big]. \tag{9}$$

Now, we analyze overestimation bias, similar to the work of Chen et al. (2021b) and Lan et al. (2020) but in the soft learning framework instead of discrete actions. Our main purpose is to find the source of critic overestimation and to devise a pessimistic Bellman operator to prevent overestimation.

**Definition 4.1.** *A random variable $X \in \mathbb{R}$ with mean $\mu = \mathbb{E}[X]$ is called sub-Gaussian with variance proxy $\sigma^2$ if its moment generating function satisfies*

$$\mathbb{E}[\exp(\lambda X)] \le \exp\big(\lambda\mu + \frac{1}{2}\lambda^2\sigma^2\big), \quad \forall \lambda \in \mathbb{R}. \tag{10}$$

Let $\mu(s,a) = \mathbb{E}_{Q \sim \mathcal{Q}}[Q(s,a)]$. We define overestimation error as difference between $\mathbb{E}_{Q \sim \mathcal{Q}}[\mathcal{T}^*Q(s,a)]$ and average $\mathcal{T}^*\mu$ as $\epsilon$,

$$\epsilon(s,a) = \mathbb{E}_{Q \sim \mathcal{Q}}[\mathcal{T}^*Q(s,a)] - \mathcal{T}^*\mu(s,a). \tag{11}$$

In the ideal case, $\epsilon(s,a)$ should be zero if there is no overestimation, which is not the case due to critic uncertainty. To quantify it, we assume that critic distribution $\mathcal{Q}(s,a)$ is sub-Gaussian with variance proxy $\sigma^2(s,a)$, representing uncertainty. If there exist an upper bound for overestimation, this bound can be used to devise a conservative Bellman backup operator. For this, we present Theorem 4.1. The proof is available in Appendix A.

**Theorem 4.1** (Overestimation quantification for sub-Gaussian critics). *Given estimated critic distribution $\mathcal{Q} \in \mathcal{P}(\mathbb{R}^{\mathcal{S} \times \mathcal{A}})$, let $\mathcal{Q}(s,a)$ be sub-Gaussian with mean $\mu(s,a)$ and variance proxy $\sigma^2(s,a)$ for all state-action pairs, with bounded support. Then,*

$$\mathbb{E}_{Q \sim \mathcal{Q}}[\mathcal{T}^*Q(s,a)] \le R(s,a) + \gamma \mathbb{E}_{s' \sim \tau(\cdot|s,a)} \Big[ \alpha \log \Big( \int_{\mathcal{A}} \exp(\alpha^{-1}\mu(s',a') + \frac{1}{2}\alpha^{-2}\sigma^2(s',a'))da' \Big) \Big]. \tag{12}$$

*In addition, overestimation due to uncertainty of estimated distribution $\mathcal{Q}$, denoted as $\epsilon$, is upper bounded for Bellman updates,*

$$\epsilon(s,a) \le \frac{\gamma}{2\alpha} \mathbb{E}_{s' \sim \tau(\cdot|s,a)} \Big[ \max_{a'} \sigma^2(s',a') \Big]. \tag{13}$$

**Corollary 4.1.1** (Pessimistic critic target). *Given estimated critic distribution $\mathcal{Q}$, using shifted distribution $\tilde{\mathcal{Q}} = \mathcal{N}(\tilde{\mu}, \tilde{\sigma}^2)$ for Bellman updates, where mean is shifted $\tilde{\mu} = \mu - \beta\sigma$ with same variance proxy $\tilde{\sigma}^2 = \sigma^2$, prevents overestimation as long as $\beta \ge \max_{(s',a')} \frac{1}{2}\alpha^{-1}\sigma(s',a')$.*

The source of overestimation and necessity of pessimistic training is revealed in Theorem 4.1 and Corollary 4.1.1. Although using pessimistic critic targets for critic and policy training is not a new idea (Moskovitz et al., 2021; Kuznetsov et al., 2020; Chen et al., 2021b), the question of how to determine pessimism ($\beta$) remains. For this, Moskovitz et al. (2021) had shown that optimal pessimism/optimism depends on the environment and learning method. Another question is about the determination of predictive critic distribution. For overestimation mitigation, we believe that using a distributional critic network is enough, which mainly models aleatoric uncertainty.

# 5 Stochastic Actor-Critic

In this section, we discuss key mechanisms needed for computation and sample efficient actor-critic learning and propose our algorithm *Stochastic Actor-Critic*. This algorithm employs *single distributional* critic network which captures aleatoric uncertainty and Bayesian dropout for epistemic uncertainty instead of ensembling. Unlike other methods, STAC uses critic estimates in a pessimistic manner using only aleatoric uncertainty, for policy evaluation and policy improvement.

The rationale behind this argument is that most of the TD target randomness due to ongoing policy updates are inherently appears as aleatoric uncertainty. Overestimation is caused by this randomness due to critic optimization process. Moreover, out-of-distribution samples are detected by high epistemic uncertainty, and should not be disregarded by pessimism. On the contrary, such samples should be explored using suitable methods. Therefore, epistemic uncertainty should not be used for overestimation mitigation in our opinion.

## 5.1 Distributional (Heteroscedastic) Critic

Distributional (heteroscedastic) networks output probability distribution instead of a point estimate and are designed to model aleatoric uncertainty of underlying phenomena (Kendall & Gal, 2017; Lakshminarayanan et al., 2017). In addition to this property, modeling output as a distribution allows the network to learn loss attenuation and makes learning robust to noisy data (Kendall & Gal, 2017). In our setting, the objective is to fit a distribution of Bellman backup uncertainty which is sourced by non-stationarity of learning procedure (ongoing policy changes and noise due to optimization process) (Dabney et al., 2021), uncertainty due to difficulty of assessing a value on some parts of state-action space (limited model capacity), and stochasticity of state transition if exists. The first two mentioned sources of uncertainty are the main reasons for overestimation, and this uncertainty modeled by the distributional critic can be used for pessimistic updates of the critic itself and the policy.

For simplicity, STAC models critic as normal distribution. This contradicts with bounded distribution assumption of Theorem 4.1, but it yields a simple loss function and is easy to interpret. Still, it is reasonable to assume critic distribution is bounded for finite horizon or discounted MDPs ($\gamma < 1$) with bounded reward functions.

Another important note is that STAC models aleatoric uncertainty for only one step, i.e., the mean of next-state value is bootstrapped instead of a random sample, since we are interested in uncertainty raised by function approximation errors rather than cumulative return distribution, unlike previous distributional RL methods. However, errors due to critic overestimation and environment/reward stochasticity still blends in a single normal distribution. Therefore, pessimism should be carefully selected (should not be too far from zero) in highly stochastic environments, since high pessimism may yiled to focus on less stochastic parts of the environment.

## 5.2 Pessimistic Objective

Like most algorithms, the natural way to inhibit overestimation is by employing pessimistic critic updates. Given that critic value distribution is normal (still sub-Gaussian), we can use modified pessimistic distribution $\tilde{\mathcal{Q}} = \mathcal{N}(\mu - \beta\sigma, \sigma^2)$ from Corollary 4.1.1, but it would be overpessimistic for higher $\beta$ values which are required to guarantee overestimation prevention.

According to Corollary 4.1.1, pessimistic TD targets should be used to train critic network. However, this analysis do not account for policy learning since policy is assumed as softmax over action values. In actor-crtic framework, same pessimistic objective should be used for policy improvement, since policy evaluation objective and policy improvement objective should be the same. In other words, at learning step $k$, Bellman backup must be equal to Bellman policy evaluation with the new policy, $\mathcal{T}^*Q^k = \mathcal{T}^{\pi^{k+1}}Q^k$. It is only possible by using same pessimistic critic value for policy improvement.

There are many factors affecting the optimal pessimism level. For example, policy improvement is slower than policy evaluation in actor-critic methods, decreasing the degree of overestimation. In addition, the real variance might be lower than the estimated variance. Lastly, overestimation may not even occur as much as the bound. For this purpose, we state that $\beta$ simply stands as a pessimism parameter to be tuned for each environment and learning hyper-parameters and it can be small depending on the learning process. At the end, we define the pessimistic expected Bellman update $\mathbb{E}_{Q \sim \tilde{\mathcal{Q}}}[\mathcal{T}^*Q(s,a)]$ as follows;

$$\mathbb{E}_{Q \sim \tilde{\mathcal{Q}}}[\mathcal{T}^*Q(s,a)] = R(s,a) + \gamma \mathbb{E}_{\substack{Q \sim \tilde{\mathcal{Q}} \\ s' \sim \tau(\cdot|s,a)}} \left[ \alpha \log \Big( \int_{\mathcal{A}} \exp(\alpha^{-1}Q(s',a'))da' \Big) \right]. \tag{14}$$

Normal distribution assumption yields the tightest bound for overestimation, converting second inequality into an equality in the proof of Theorem 4.1. This way, overestimation is closer to the upper bound and yields the worst case.

There is no analytical way to determine the optimum pessimism level. Moskovitz et al. (2021) focus on updating pessimism *on the fly* as a bandit problem instead of fixing it but this requires evaluating on-policy returns and introduces an online bandit to update pessimism. This approach would work in an off-policy online setting surely, but it is not usable in a completely offline setting since there would be no feedback for the bandit. To allow STAC to be also used in offline settings in the future, the pessimism level is defined as a fixed hyper-parameter.

## 5.3 Dropout Regularization

Dropout regularization (Srivastava et al., 2014) allows to capture the probabilistic nature of a network, representing Bayesian neural networks (Gal & Ghahramani, 2016). It is also equivalent to representing the model as an ensemble since each sampled weight set of the network corresponds to a sub-model (He et al., 2021). For this purpose, STAC employs dropout regularization for both critic and policy networks. Neural architectures of critic and policy are illustrated in Appendix D.

Epistemic uncertainty modeled by dropout is not used for pessimistic learning. It has an effect if a given state-action pair is out-of-distribution and pessimism would hinder the agent from exploring these states and actions. However, aleatoric uncertainty would be high on in-distribution but stochastic parts of state-action space, which are not related to exploration.

Dropout regularizes the learning procedure, and promotes exploration in a heuristic way. In STAC, dropout is active in all phases of learning. Each forward pass randomly draws an action to take, and a value estimate for policy improvement in an epistemic manner, yielding Thompson sampling for exploration (Gal et al., 2016a; 2017).

## 5.4 Layer Normalization

Layer Normalization (Ba et al., 2016) is a normalization method applied to feature dimensions of activations. It has a regularization effect and prevents possible numerical instabilities in training time. In STAC, we implement Layer Normalization after all hidden activations of critic and policy networks, similar to Hiraoka et al. (2021).

### 5.5 Algorithm Summary

Finally, we present the Stochastic Actor-Critic (STAC) algorithm using the results of analyses from previous sections. Unlike previous methods, we parameterize critic $\mathcal{Q}_\theta$ as a single network by parameter set $\theta$ and policy $\pi_\phi$ as another single network by parameter set $\phi$, where both networks have probability distribution as outputs, that is, networks represent distributions over values and actions. Policy outputs a `tanh` transformed normal distribution to bound actions to $[-1, 1]$. Networks illustrations are available in Appendix D. Note that the bar notation stands for the lagged network with non-trainable parameters. STAC is summarized in Algorithm 1 with gradient descent but Adam optimizer (Kingma & Ba, 2014) is used in our experiments.

**Critic learning (policy evaluation)**  Critic network predicts cumulative return with some uncertainty. Using transition tuples from experience replay as batch, $\mathcal{D}_b = \{(s_i, a_i, r_i, s_i', \texttt{done}_i)\}_{i=1}^{N_b}$, temporal difference (TD) target $Q_i^{TD}$, representing Bellman backup, is $\beta$-pessimistic,

$$Q_i^{TD} = r_i + \gamma(\mu_{\bar{\theta}}(s_i', \tilde{a}_i') - \beta\sigma_{\bar{\theta}}(s_i', \tilde{a}_i') - \alpha\log\pi_\phi(s_i', \tilde{a}_i'))(\neg\texttt{done}_i), \quad \tilde{a}_i' \sim \pi_\phi(\cdot \mid s_i'). \tag{15}$$

Learning objective is cross-entropy loss (log loss),

$$\mathcal{L}_\theta(\mathcal{D}_b) = \frac{1}{N_b} \sum_{i=1}^{N_b} -\log\mathcal{Q}_\theta(Q_i^{TD} \mid s_i, a_i). \tag{16}$$

Theoretically, critic distribution is not restricted to any type but sub-Gaussian. For simplicity, we model the critic to be represented as a normal distribution, i.e. $\mathcal{Q}_\theta = \mathcal{N}(\mu_\theta, \sigma_\theta^2)$ in this work. In this case, the cross entropy loss becomes as follows;

$$\mathcal{L}_\theta(\mathcal{D}_b) = \frac{1}{2}\log 2\pi + \frac{1}{N_b}\sum_{i=1}^{N_b}\left(\frac{1}{2}\log\sigma_\theta^2(s_i, a_i) + \frac{(Q_i^{TD} - \mu_\theta(s_i, a_i))^2}{2\sigma_\theta^2(s_i, a_i)}\right). \tag{17}$$

**Lagged critic for TD target**  When the trained critic network is also used in calculating the target value, the critic training is prone to divergence (Li et al., 2023b). For this, a common approach is to use another critic network to evaluate TD target (Mnih et al., 2013). Similar to Lillicrap et al. (2015), Fujimoto et al. (2018), and Haarnoja et al. (2018), we use a delayed form of critic network for TD target evaluations as demonstrated in Equation 15. The parameters of target critic are only updated by Polyak averaging of main critic network weights through learning steps; $\bar{\theta} \leftarrow \rho\bar{\theta} + (1 - \rho)\theta$. This strategy is important to ensure the stability of temporal difference learning.

**Policy improvement**  The policy improvement objective has a very similar form to SAC algorithm (Haarnoja et al., 2018) except using standard deviation to construct $\beta$-pessimistic objective. Pessimism is also here to be consistent with pessimistic Bellman backup definition. Using states only from experience replay as batches $\mathcal{D}_b = \{(s_i)\}_{i=1}^{N_b}$ with batch size $N_b$, loss function for policy network is as follows;

$$\mathcal{L}_\phi(\mathcal{D}_b) = \frac{1}{N_b}\sum_{i=1}^{N_b}\mathbb{E}_{a\sim\pi_\phi(\cdot\mid s_i)}\big[\mu_\theta(s_i, a) - \beta\sigma_\theta(s_i, a) - \alpha\log\pi_\phi(a \mid s_i)\big]. \tag{18}$$

**Automatic temperature tuning**  Using constant temperature results in different policies if the reward magnitude changes. To mitigate this, Haarnoja et al. (2018) proposed a policy entropy constraint, representing temperature as the Lagrange multiplier of the constraint. Given target entropy $\bar{\mathcal{H}}$ as hyper-parameter, the loss function related to this constraint is as follows;

$$\mathcal{L}_\alpha(\mathcal{D}_b) = -\alpha\bar{\mathcal{H}} + \alpha\sum_{i=1}^{N_b}\mathbb{E}_{a\sim\pi_\phi(\cdot\mid s_i)}\big[-\log\pi_\phi(a \mid s_i)\big]. \tag{19}$$

---

**Algorithm 1** Stochastic Actor-Critic

---

**Require:** Environment `env`
**Require:** Experience buffer $\mathcal{D}$
**Require:** Critic $\mathcal{Q}_\theta$, lagged critic $\mathcal{Q}_{\bar{\theta}}$, policy $\pi_\phi$, all with dropout
**Require:** Initial temperature $\alpha$, target entropy $\bar{\mathcal{H}}$
**Require:** Pessimism $\beta$
**Require:** Learning rates $\eta_Q$, $\eta_\pi$, $\eta_\alpha$, Polyak parameter $\rho$
**Require:** Total training steps $N$, batch size $N_b$

   $s \sim$ `env.reset()`                 $\triangleright$ Reset the environment
   **for** $N$ timesteps **do**
      $a \sim \pi_\phi(\cdot \mid s)$                 $\triangleright$ Sample action
      $r, s', $ `done` $\sim$ `env.step`$(a)$             $\triangleright$ Act on environment
      $\mathcal{D} \leftarrow \mathcal{D} \cup (s, a, r, s', $ `done`$)$        $\triangleright$ Record transition tuple
      **if** `done` **then** $s \leftarrow s'$ **else** $s \sim$ `env.reset()`       $\triangleright$ State transition or reset
      **for** $G$ gradient steps **do**
         $\mathcal{D}_b = \{(s_i, a_i, r_i, s'_i, $ `done`$_i)\}_{i=1}^{N_b} \sim \mathcal{D}$     $\triangleright$ Sample minibatch for training
         $\tilde{a}'_i \sim \pi_\phi(\cdot \mid s'_i), \quad \forall i \in \{1, 2, ..., N_b\}$     $\triangleright$ Sample next actions
         $Q_i^{TD} = r_i + \gamma(\mu_{\bar{\theta}}(s'_i, \tilde{a}'_i) - \beta\sigma_{\bar{\theta}}(s'_i, \tilde{a}'_i))(\neg$ `done`$_i), \quad \forall i \in \{1, 2, ..., N_b\}$    $\triangleright$ Build TD targets
         $\theta \leftarrow \theta - \eta_Q \nabla_\theta \left( \frac{1}{N_b} \sum_{i=1}^{N_b} -\log \mathcal{Q}_\theta(Q_i^{TD} \mid s_i, a_i) \right)$    $\triangleright$ Update critic
         $\phi \leftarrow \phi - \eta_\pi \nabla_\phi \left( \frac{1}{N_b} \sum_{i=1}^{N_b} \mathbb{E}_{a \sim \pi_\phi(\cdot \mid s_i)} \left[ \mu_\theta(s_i, a) - \beta\sigma_\theta(s_i, a) - \alpha \log \pi_\phi(a \mid s_i) \right] \right)$   $\triangleright$ Update policy
         $\alpha \leftarrow \alpha - \eta_\alpha \nabla_\alpha \left( -\alpha\bar{\mathcal{H}} + \alpha \sum_{i=1}^{N_b} \mathbb{E}_{a \sim \pi_\phi(\cdot \mid s_i)} \left[ -\log \pi_\phi(a \mid s_i) \right] \right)$   $\triangleright$ Update temperature
         $\bar{\theta} \leftarrow \rho\bar{\theta} + (1 - \rho)\theta$         $\triangleright$ Update target critic network
      **end for**
   **end for**

---

## 6 Experiments

Our experiments aim to investigate whether enhancing off-policy actor-critic methodology with STAC can improve their sample and computation efficiency on difficult continuous-control benchmarks. For this purpose, STAC is compared to similar competitive algorithms; TQC (Kuznetsov et al., 2020), DROQ (Hiraoka et al., 2021), SAC (Haarnoja et al., 2018) and TOPSAC, which is SAC variant of TOP algorithm (Moskovitz et al., 2021), where only exploration scheme is changed to maximum entropy policy. We also run REDQ Chen et al. (2021b) with UTD ratio ($G$) equal to 1, but results were very similar to the SAC, so results are not demonstrated not to overcrowd figures and tables. All algorithm results are obtained using in-house code with the same network architectures (including layer normalization) to make a fair comparison. We included DROQ algorithm with UTD ratio ($G$) equal to 1 and 5, although it is equal to 20 in the original paper.

Ablation studies are conducted to examine the effectiveness of different levels of pessimism under varying dropout rates, and the effect of dropout under fixed pessimism. The effect of Layer Normalization is not surveyed since it is done by Hiraoka et al. (2021) extensively for DROQ algorithm.

Through Gymnasium API (Towers et al., 2023), six MuJoCo are used for comparison as they are tested by most algorithms in the literature. To assess performance on stochastic environments, `BipedalWalker-v3` and `BipedalWalkerHardcore-v3` from Box2D are also tested as the terrain where the walker walks is randomly generated, shown in Figure 3 and 4. Hyper-parameters per environment can be found in Table 4 of Appendix C. For all experiments, PyTorch (version 2.2.2) (Paszke et al., 2019) is used. Please refer to Appendix E for the codebase.

**Evaluation protocol** After each 1000 time steps, we execute a single test episode using the online policy and measure its performance by calculating the total reward accumulated during the episode. Total training steps are 50k for `InvertedDoublePendulum-v4` and 300k for the rest.

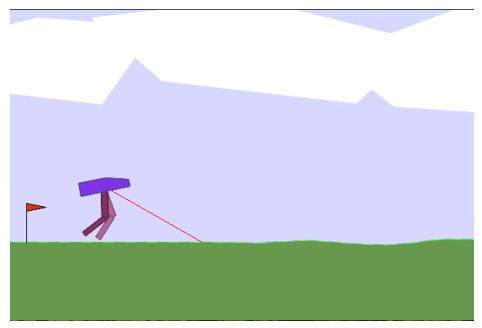

Figure 3: BipedalWalker-v3 Environment.

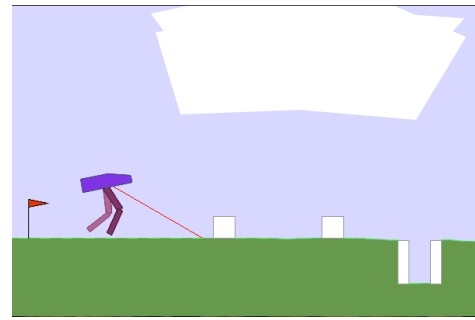

Figure 4: BipedalWalkerHardcore-v3 Environment.

**Learning curves** Specified environments are trained through a fixed number of environment interactions, repeated 5 times to assess the stability of the algorithm shown by mean and standard deviation. In Figure 5, the performance of STAC is shown against previously mentioned SOTA algorithms for 8 tasks, where important hyper-parameters yielding best results are used for STAC and TQC, summarized in Table 5. Additionally, value estimation errors are presented in Figure 6. The bold lines represent the average, while the shaded area indicates the standard deviation (to represent randomness through seeds) of the total reward across evaluation episodes. Further experimental details are presented in Appendix C. Mean and standard deviation of episodic returns over five training runs are summarized in Table 1. Average returns through all learning processes averaged over random seeds are summarized in Table 2.

Table 1: Episodic return over five training runs on MuJoCo tasks at the end of training. ± sign denotes one standard deviation across trials. The best method scores are highlighted bold.

| Env | # steps | DROQ G=1 | DROQ G=5 | SAC | STAC | TOPSAC | TQC |
|---|---|---|---|---|---|---|---|
| Ant-v4 | 300k | 1227±1058 | 2800±1738 | 3074±1306 | **4646±1422** | 1544±1399 | 4144±2432 |
| BipedalWalker-v3 | 300k | 151.29±187.49 | 224.29±130.31 | 199.67±144.71 | 284.42±65.86 | 304.90±82.32 | **323.41±41.17** |
| BipedalWalkerHardcore-v3 | 300k | -60.28±39.19 | -41.20±81.15 | -93.54±31.51 | **16.62±99.57** | 8.04±91.74 | -7.18±84.09 |
| HalfCheetah-v4 | 300k | 6655±996 | 7519±841 | 7285±714 | 8084±1501 | 7615±1143 | **8739±725** |
| Hopper-v4 | 300k | 1655±975 | 1568±976 | 1675±1113 | **2649±1059** | 1197±819 | 2084±1308 |
| Humanoid-v4 | 300k | 1616±1230 | 1794±1425 | 1788±1228 | **4540±1715** | 1480±1578 | 2674±2047 |
| InvertedDoublePendulum-v4 | 50k | 9022±1435 | 8919±1881 | 8286±2550 | **9358±1** | 7735±3248 | 8658±2169 |
| Walker2d-v4 | 300k | 2655±1574 | 1119±899 | 1869±1599 | **4915±132** | 2013±1270 | 4219±845 |

Table 2: Average episodic return through learning procedure and over five training runs on MuJoCo and Box2D tasks. The best method scores are highlighted bold.

| Env | # steps | DROQ G=1 | DROQ G=5 | SAC | STAC | TOPSAC | TQC |
|---|---|---|---|---|---|---|---|
| Ant-v4 | 300k | 983.09 | 2046.47 | 1582.81 | 2742.97 | 1140.96 | **3056.95** |
| BipedalWalker-v3 | 300k | 51.95 | 146.52 | 79.35 | 229.16 | 218.89 | **242.11** |
| BipedalWalkerHardcore-v3 | 300k | -78.93 | -71.18 | -83.24 | **-23.51** | -32.25 | -35.59 |
| HalfCheetah-v4 | 300k | 5323.60 | 6093.85 | 5751.63 | 6309.67 | 5580.58 | **6524.20** |
| Hopper-v4 | 300k | 1154.54 | 614.65 | 1029.88 | **1956.80** | 1023.92 | 1954.46 |
| Humanoid-v4 | 300k | 952.86 | 994.21 | 1182.68 | **2271.19** | 833.41 | 1880.35 |
| InvertedDoublePendulum-v4 | 50k | 5967.47 | **6812.87** | 5578.02 | 6592.03 | 5749.87 | 5624.03 |
| Walker2d-v4 | 300k | 1490.90 | 1300.85 | 1371.76 | **3166.30** | 1822.86 | 2963.88 |

**Sample efficiency** As seen from Figure 5 and Table 1, STAC outperforms other algorithms, except TQC for some of the environments, in terms of sample efficiency. The main explanation is in Figure 6, as other algorithms except TQC suffer from positive overestimation bias, where STAC handles it by using a pessimism level specifically selected for each environment. It is also the same for TQC algorithm, as we found the number of quantiles to drop per network by trial-and-error to represent pessimism. Also as seen in Table 2, STAC also performs well not only at the end of training but also during whole learning time along with TQC.

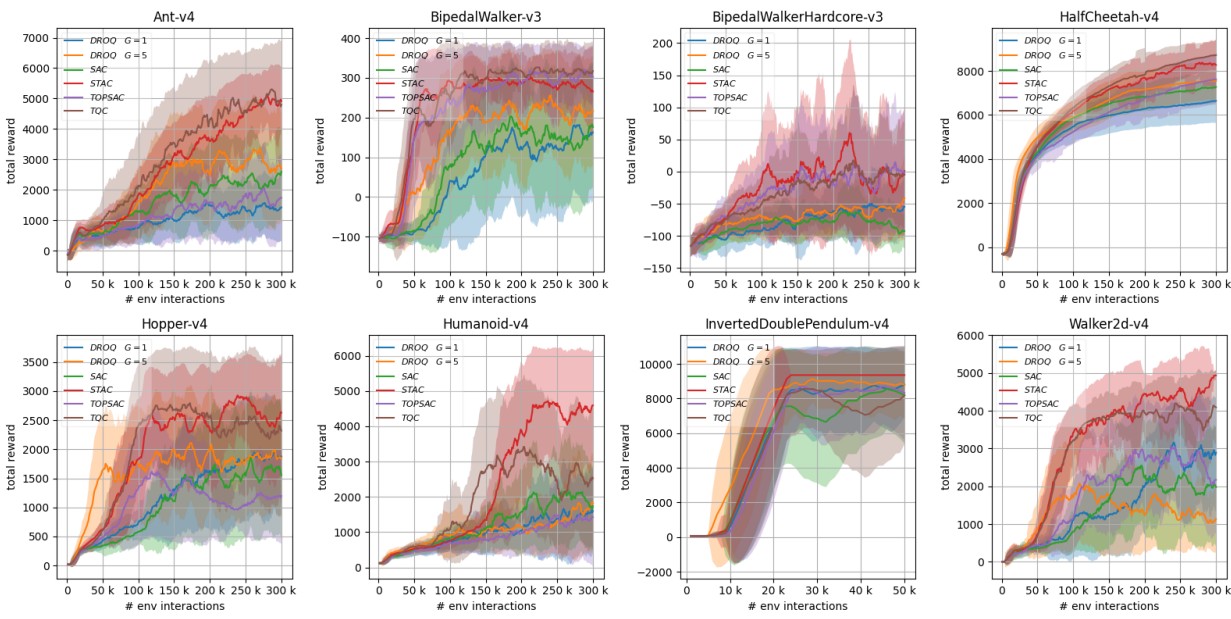

Figure 5: Main learning curves of STAC and other algorithms. The standard deviation is represented by the shaded areas, while the average return across evaluation episodes is shown by solid curves. See specific hyper-parameters from Table 5.

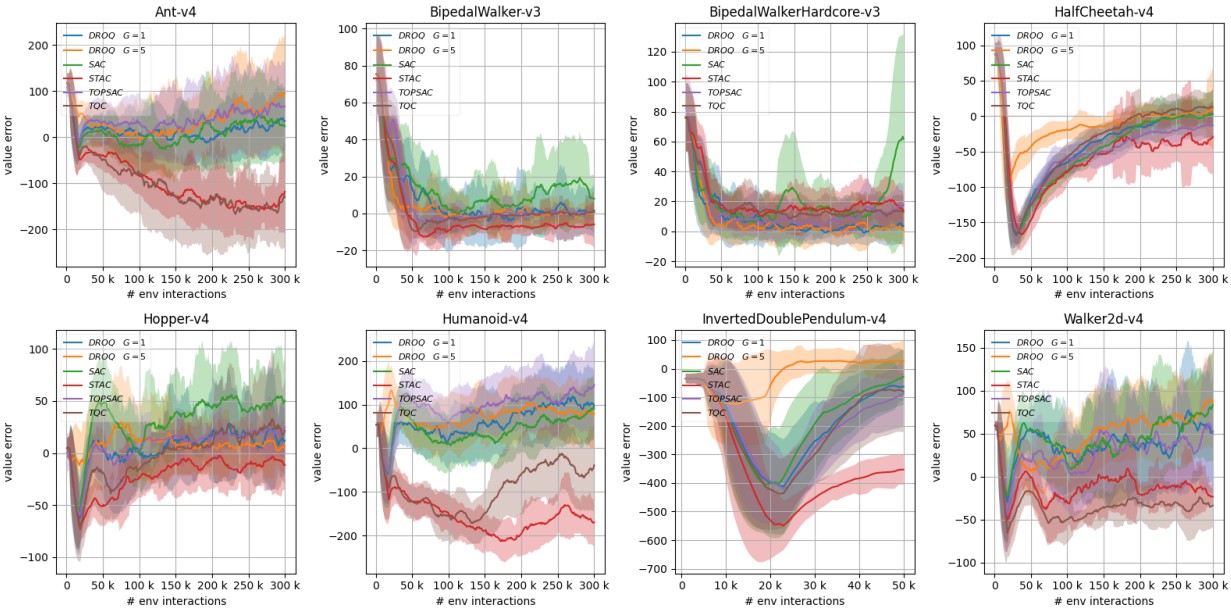

Figure 6: Estimation error of STAC and other algorithms on the beginning of episodes. The standard deviation is represented by the shaded areas, while the average errors across evaluation episodes are shown by solid curves. See specific hyper-parameters from Table 5.

**Computation efficiency**  As wall-clock time statistics vary depending on computing units and environmental conditions, we present the number of critic networks, number of critic backpropagations, and number of target critic calls per time step in Table 3. Note that all methods use a single and same policy network architecture, and only the input and output layers of the critic are different which has an insignificant effect on the number of training parameters. Although TQC performs slightly better than STAC in terms of sample efficiency for some environments, STAC uses fewer parameters and consumes fewer computation resources compared to TQC since it employs only a single critic network with a UTD ratio of 1, outperforming other algorithms in terms of computation efficiency.

Table 3: Number of critic networks and backprops per time step. Each critic has same hidden activation size.

|  | STAC | DROQ G=1 | DROQ G=5 | SAC | TOPSAC | TQC |
|---|---|---|---|---|---|---|
| # critic network | 1 | 2 | 2 | 2 | 2 | 5 |
| # critic backprop | 1 | 2 | 10 | 2 | 2 | 5 |
| # target critic call | 1 | 2 | 10 | 2 | 2 | 5 |

However, the key behind STAC's performance depends on two main hyper-parameters, pessimism and dropout rate of critic and policy networks. In order to understand effects and algorithm sensitivity to those parameters, pessimism sweeps are conducted by varying dropout. In addition, same sweep is conducted by turning off dropout for target critic network while main critic network still has a dropout rate of 0.01, to see the effect of not bootstrapping dropout uncertainty. Lastly, using same pessimism level, effect of dropout is summarized and discussed.

## 6.1   Pessimism Sweep with varying Dropout Rates

To investigate the sensitivity of STAC to pessimism parameter $\beta$, we run STAC on all environments by varying $\beta$, for 3 different dropout rates and turned off target critic dropout. Learning curves are available in Appendix B.1. Results for 0.01 dropout, as shown in Figure 7, indicate that $\beta$ is a sensitive parameter. In Figure 8, the higher $\beta$ yields a higher negative error, consistent with our assumptions. The effect of pessimism is also similar for different dropout configurations, as shown in Figure 9, 11 and 13. Therefore, pessimism should be determined carefully to guarantee better performance. Excess pessimism paves the way to underestimation, whereas lack of it causes critic overestimation which is inherent to actor-critic methods.

In addition, score curves are worse if value estimations tend to be positive (Figure 8, 10, 12 and 14), meaning that critic overestimation is not mitigated enough (see `BipedalWalker-v3`, `Hopper-v4`, `Humanoid-v4`, `InvertedDoublePendulum-v4`). On the other hand, score curves are again worse when error curves are negative and far from zero, meaning that the learner is stuck on critic underestimation caused by high pessimism (see `Ant-v4`, `BipedalWalkerHardcore-v3`, `HalfCheetah-v4`, `Walker2d-v4`). In the end, pessimism sensitivity varies for different environments, possibly because of varying task difficulties. For easier tasks, less pessimism is enough but difficult tasks require significant pessimism. This parameter stands as the major bottleneck of STAC and can only be determined by this heuristic for now.

**Pessimism under Environment Stochasticity**  Pessimistic learning on aleatoric uncertainty may hinder exploration, especially in environments with high stochasticity and sparse rewards. We compared results from `BipedalWalker-v3` and `BipedalWalkerHardcore-v3`. The second one is harder because it has more obstacles, making rewards sparse and transitions more unpredictable (see Figure 3 and 4). To eliminate the dropout effect, let us look at Figure 11: in the harder environment, less pessimism works better. In simpler setting (`BipedalWalker-v3`), most aleatoric uncertainty comes from model errors, so some pessimism helps overestimation mitigation. In more stochastic setting (`BipedalWalkerHardcore-v3`), aleatoric uncertainty also comes from the environment itself. If overestimation is already less, too much pessimism may make the agent overly cautious, limiting exploration. Since the goal in reinforcement learning is to maximize the average return, not the best or worst return, the agent should stay neutral in response to environment-driven aleatoric uncertainty. In short, pessimism is useful in stable environments to correct overestimation, but in more stochastic settings, it is better to be near-neutral but not optimistic upon aleatoric uncertainty.

## 6.2 Dropout Effect

Dropout is also an important parameter as it determines epistemic uncertainty and regularizes learning. Learning curves are available in Appendix B.2. For this ablation, previously run experiments are combined with the best performing $\beta$ parameter. As it can be seen from Figure 15, best dropout rate varies for each environment. This is the reason behind using different dropout for the mentioned environments in the main comparison study (see Table 5).

The optimal dropout rate varies depending on the task, and it is even zero for some of the environments. Although using dropout promotes exploration, it also regularizes learning. Therefore, it possibly causes under-exploitation/over-exploration for some environments (Gal & Ghahramani, 2016). STAC is also tested with turned off target critic dropout, to understand the effect of learning epistemic uncertainty sourced by dropout within the distributional representation. For most environments, it also works with slight performance loss except `Humanoid-v4`, although learning well at the beginning. Value estimation error increases in the positive direction and this error is even more than the zero dropout case, as shown in Figure 16. We believe that this is caused by mismatch between trained and target critic since trained critic still employs dropout.

## 7 Conclusion & Future Directions

In this paper, we introduced Stochastic Actor-Critic (STAC), a novel off-policy actor-critic algorithm. The main idea is to mitigate overestimation for the sake of faster and more robust learning by incorporating the pessimistic learning objective using aleatoric uncertainty. For this, critic is modeled as a distributional (heteroscedastic) neural network. Although normal distribution is used for this purpose, our analysis is valid for all sub-Gaussian critic distributions or quantile representations. We derived an upper bound for overestimation, demonstrating that an adequate level of pessimism mitigates overestimation without succumbing to underestimation, thus facilitating computation and sample-efficient learning. Lastly, Bayesian dropout is utilized for representing epistemic uncertainty, enabling robustness and exploration.

**Adaptive Pessimism** Our ablation studies demonstrate the effects of dropout rate and pessimism, revealing the sensitivity of the learning procedure to these parameters. For each specific environment and optimization method, an optimal level of pessimism and dropout exists. A promising direction for future research is to develop a grounded method to adjust the pessimism level for specific environments and agents to allow better adaptation for the learner to the environment. In addition, the sensitivity of similar algorithms to pessimism and dropout rate should be investigated in depth.

**Pessimism under Stochastic Environments** The effect of pessimism on highly stochastic environments is also an important topic for research. While mitigating overestimation, higher pessimism may lead to risk-averse behaviour in the environment. This phoenomenon should be investigated in depth in future works. A better exploration strategy upon epistemic uncertainty may be a solution for this case.

**Exploration** Grounded methods using *optimism in the face of uncertainty* principle upon epistemic uncertainty is worth investigating, keeping pessimism upon aleatoric uncertainty. For this, different methods for modeling epistemic uncertainty other than ensembles and Bayesian dropout can be considered. Concrete dropout (Gal et al., 2017), and evidential deep learning (Sensoy et al., 2018; Amini et al., 2020) frameworks may offer better alternatives.

**Broader Impact** STAC tackles critical challenges such as accelerating learning, improving stability, and ensuring computation efficiency. Our research not only pushes the boundaries of reinforcement learning but also promises significant implications for enhancing the safety and intelligence of robots, self-driving cars, and autonomous systems in healthcare and finance.

**Acknowledgments**

This research has received no external funding.

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

## Appendix A    Proofs

*Proof of Theorem 4.1.* Analyzing expected Bellman update $\mathbb{E}_{Q \sim \mathcal{Q}}[\mathcal{T}^*Q(s,a)]$,

$$
\mathbb{E}_{Q \sim \mathcal{Q}}[\mathcal{T}^*Q(s,a)] = R(s,a) + \gamma\mathbb{E}_{s' \sim \tau(\cdot|s,a)}\Big[\mathbb{E}_{Q \sim \mathcal{Q}}\Big[\alpha\log\Big(\int_{\mathcal{A}} \exp(\alpha^{-1}Q(s',a'))da'\Big)\Big]\Big]
$$

$$
\leq R(s,a) + \gamma\mathbb{E}_{s' \sim \tau(\cdot|s,a)}\Big[\alpha\log\Big(\mathbb{E}_{Q \sim \mathcal{Q}}\Big[\int_{\mathcal{A}} \exp(\alpha^{-1}Q(s',a'))da'\Big]\Big)\Big]
$$

$$
= R(s,a) + \gamma\mathbb{E}_{s' \sim \tau(\cdot|s,a)}\Big[\alpha\log\Big(\int_{\mathcal{A}} \mathbb{E}_{Q \sim \mathcal{Q}}\Big[\exp(\alpha^{-1}Q(s',a'))\Big]da'\Big)\Big]
$$

$$
\leq R(s,a) + \gamma\mathbb{E}_{s' \sim \tau(\cdot|s,a)}\Big[\alpha\log\Big(\int_{\mathcal{A}} \exp(\alpha^{-1}\mu(s',a') + \tfrac{1}{2}\alpha^{-2}\sigma^2(s',a'))da'\Big)\Big]
$$

$$
\leq R(s,a) + \gamma\mathbb{E}_{s' \sim \tau(\cdot|s,a)}\Big[\alpha\log\Big(\Big(\int_{\mathcal{A}} \exp(\alpha^{-1}\mu(s',a'))da'\Big) \cdot \Big(\max_{a'} \exp(\tfrac{1}{2}\alpha^{-2}\sigma^2(s',a'))\Big)\Big)\Big]
$$

$$
= R(s,a) + \gamma\mathbb{E}_{s' \sim \tau(\cdot|s,a)}\Big[\alpha\log\Big(\int_{\mathcal{A}} \exp(\alpha^{-1}\mu(s',a'))da'\Big) + \frac{1}{2\alpha}\max_{a'}\sigma^2(s',a')\Big]
$$

$$
= R(s,a) + \gamma\mathbb{E}_{s' \sim \tau(\cdot|s,a)}\Big[\alpha\log\Big(\int_{\mathcal{A}} \exp(\alpha^{-1}\mu(s',a'))da'\Big)\Big] + \frac{\gamma}{2\alpha}\mathbb{E}_{s' \sim \tau(\cdot|s,a)}\Big[\max_{a'}\sigma^2(s',a')\Big].
$$

First inequality comes from Jensen's inequality (using concave property of log function) while the following equality is a result of Tonelli's theorem. The second inequality results from the property of sub-Gaussian distribution 4.1, where the first statement of the theorem is proven. The following inequality is a result of the mean value theorem for integrals. In the last equality, the first two terms are equal to $\mathcal{T}^*\mu(s,a)$. Therefore,

$$
\epsilon(s,a) = \mathbb{E}_{Q \sim \mathcal{Q}}[\mathcal{T}^*Q(s,a)] - \mathcal{T}^*\mu(s,a) \leq \frac{\gamma}{2\alpha}\mathbb{E}_{s' \sim \tau(\cdot|s,a)}\Big[\max_{a'}\sigma^2(s',a')\Big].
$$

$\square$

*Proof of Corollary 4.1.1.* From the Theorem 4.1, we can show that

$$
\mathbb{E}_{Q \sim \mathcal{Q}}[\mathcal{T}^*Q(s,a)] \leq R(s,a) + \gamma\mathbb{E}_{s' \sim \tau(\cdot|s,a)}\Big[\alpha\log\Big(\int_{\mathcal{A}} \exp(\alpha^{-1}(\mu(s',a') - \beta\sigma(s',a') + \tfrac{1}{2}\alpha^{-1}\sigma^2(s',a')))da'\Big)\Big]
$$

$$
= R(s,a) + \gamma\mathbb{E}_{s' \sim \tau(\cdot|s,a)}\Big[\alpha\log\Big(\int_{\mathcal{A}} \exp(\alpha^{-1}\mu^{\dagger}(s',a'))da'\Big)\Big] = \mathcal{T}^*\mu^{\dagger}(s,a).
$$

where we have defined $\mu^{\dagger}(s',a') = \mu(s',a') - \beta\sigma(s',a') + \tfrac{1}{2}\alpha^{-1}\sigma^2(s',a')$. If $\beta \geq \max_{(s',a')} \tfrac{1}{2}\alpha^{-1}\sigma(s',a')$, then $\mu^{\dagger}(s',a') < \mu(s',a')$. So we can show that

$$
\mathbb{E}_{Q \sim \mathcal{Q}}[\mathcal{T}^*Q(s,a)] \leq \mathcal{T}^*\mu^{\dagger}(s,a) \leq \mathcal{T}^*\mu(s,a). \tag{20}
$$

$\square$

# Appendix B    Results of Ablation Studies

## B.1    Pessimism Sweep with varying Dropout

### B.1.1    Dropout=0.01

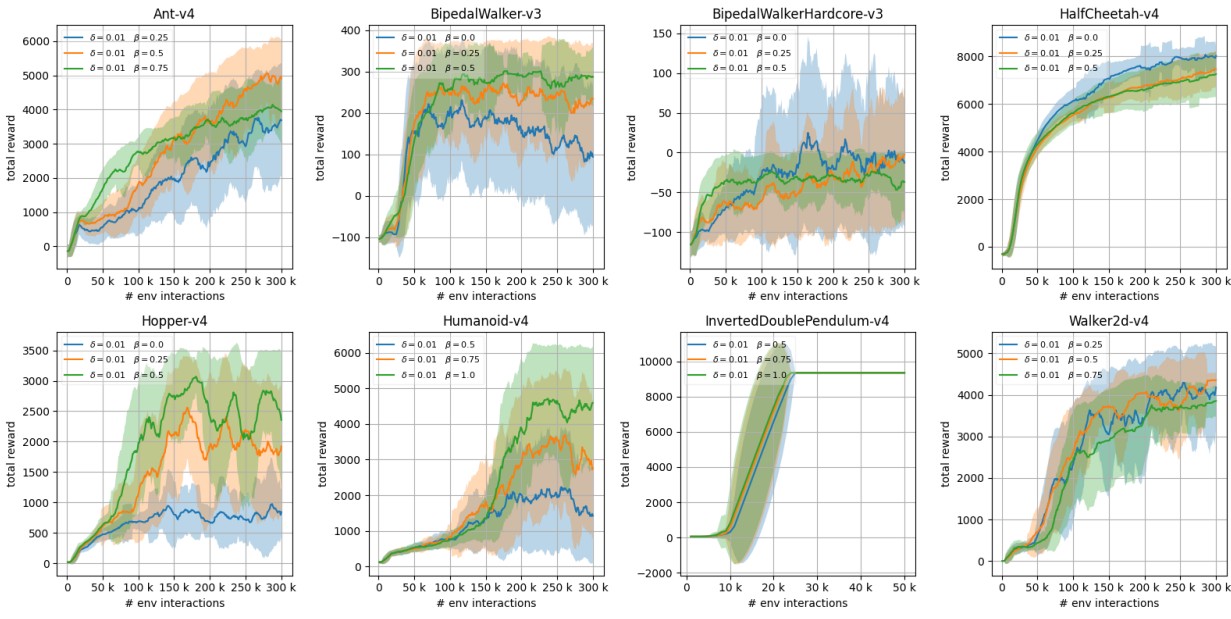

Figure 7: Learning curves of STAC with varying pessimism ($\beta$) parameter. Dropout is equal to 0.01.

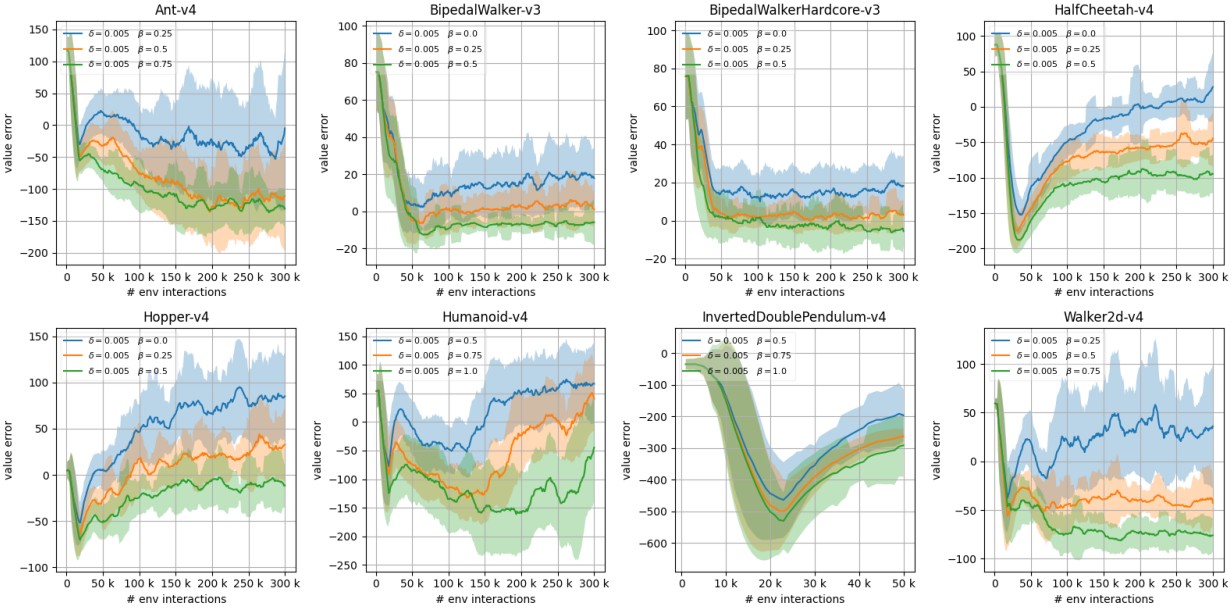

Figure 8: Episodic value estimation error curves of STAC with varying pessimism ($\beta$) parameter. Dropout is equal to 0.01.

## B.1.2 Dropout=0.005

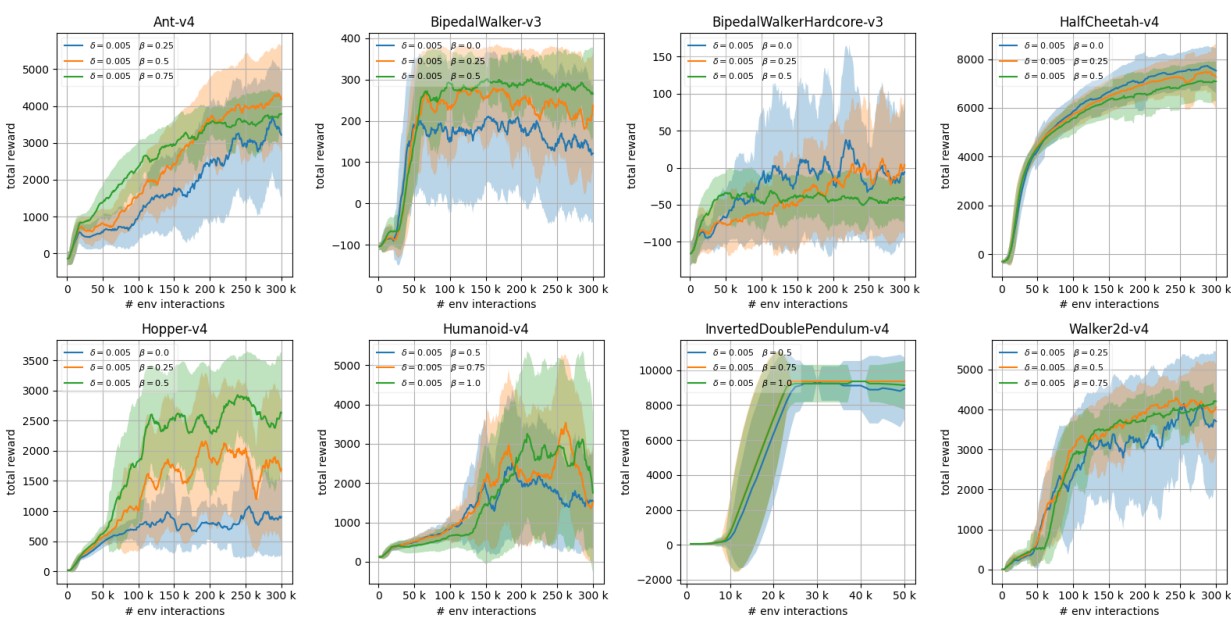

Figure 9: Learning curves of STAC with varying pessimism ($\beta$) parameter. Dropout is equal to 0.005.

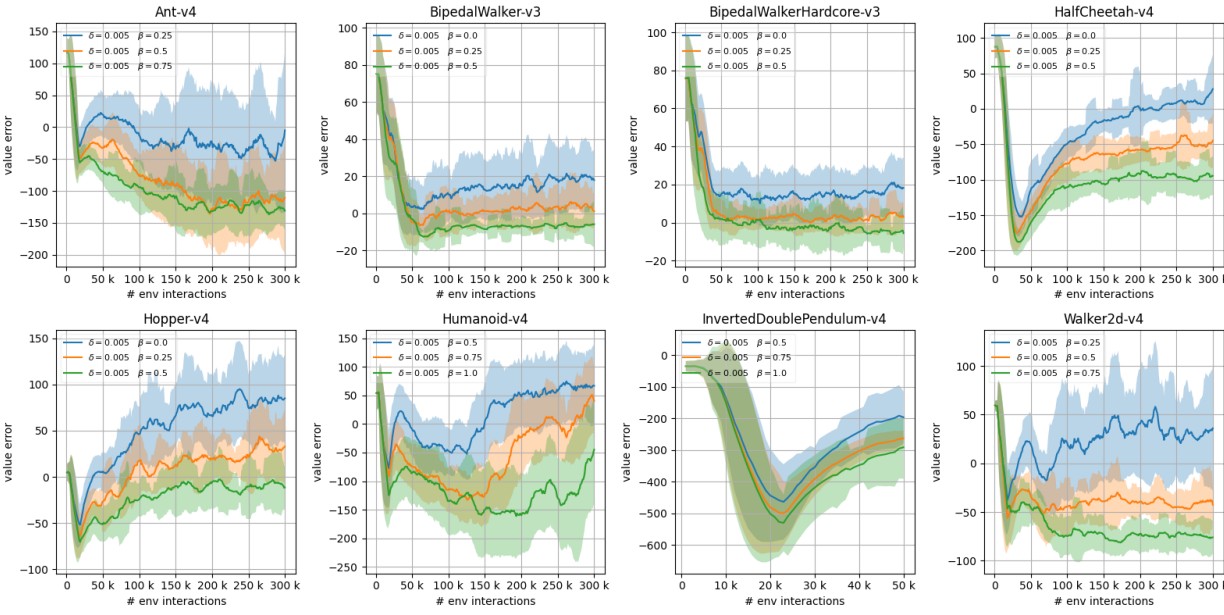

Figure 10: Episodic value estimation error curves of STAC with varying pessimism ($\beta$) parameter. Dropout is equal to 0.005.

### B.1.3   Dropout=0

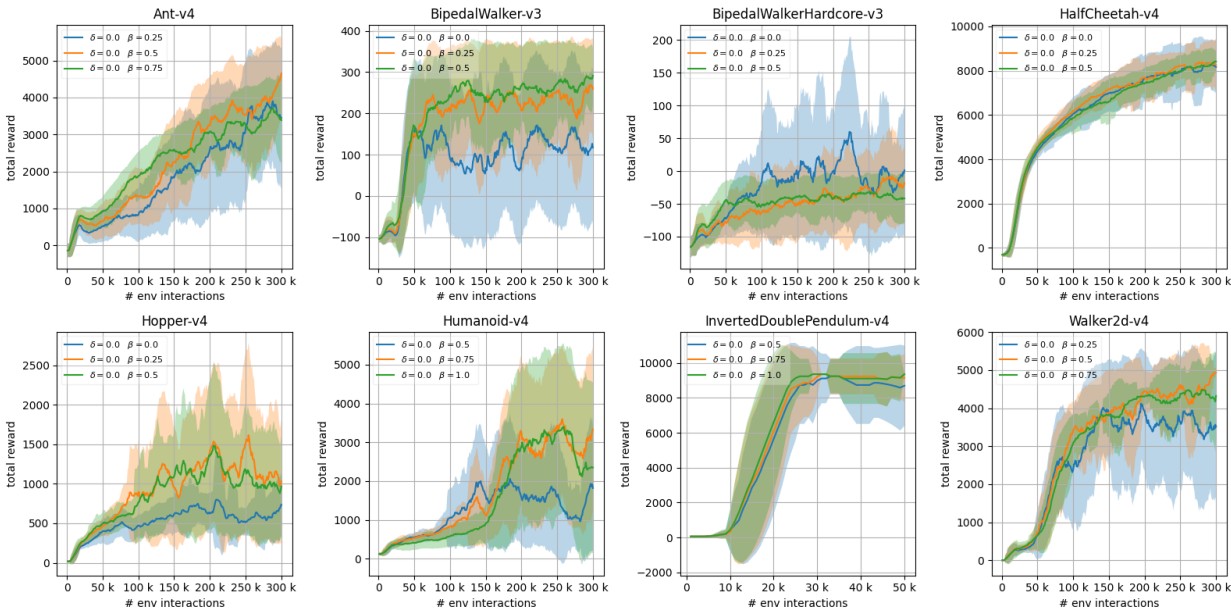

Figure 11: Learning curves of STAC with varying pessimism ($\beta$) parameter. Dropout is zero.

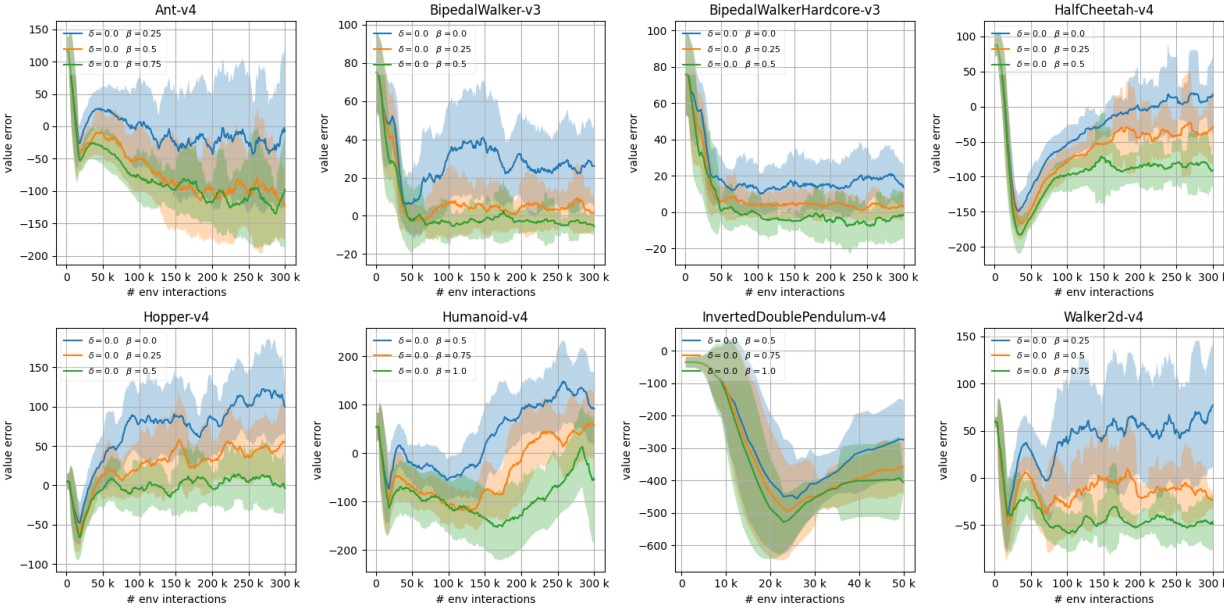

Figure 12: Episodic value estimation error curves of STAC with varying pessimism ($\beta$) parameter. Dropout is zero.

## B.1.4   Dropout=0.01, No Dropout for Target Critic

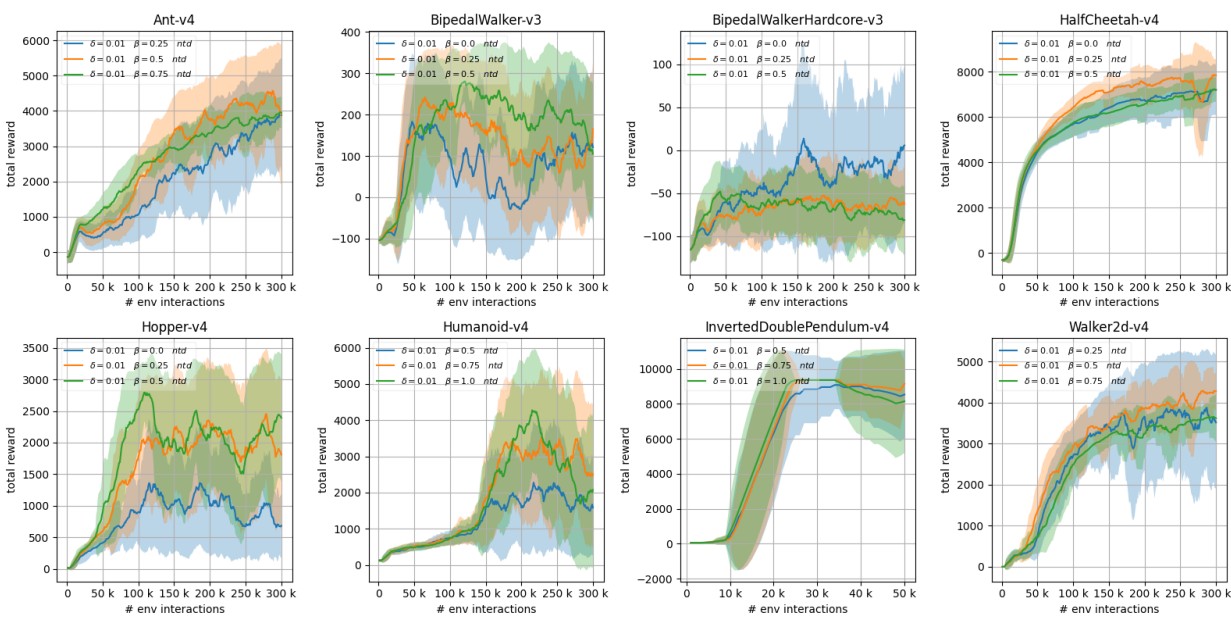

Figure 13: Learning curves of STAC with varying pessimism ($\beta$) parameter. Dropout is equal to 0.01, but target critic has no dropout.

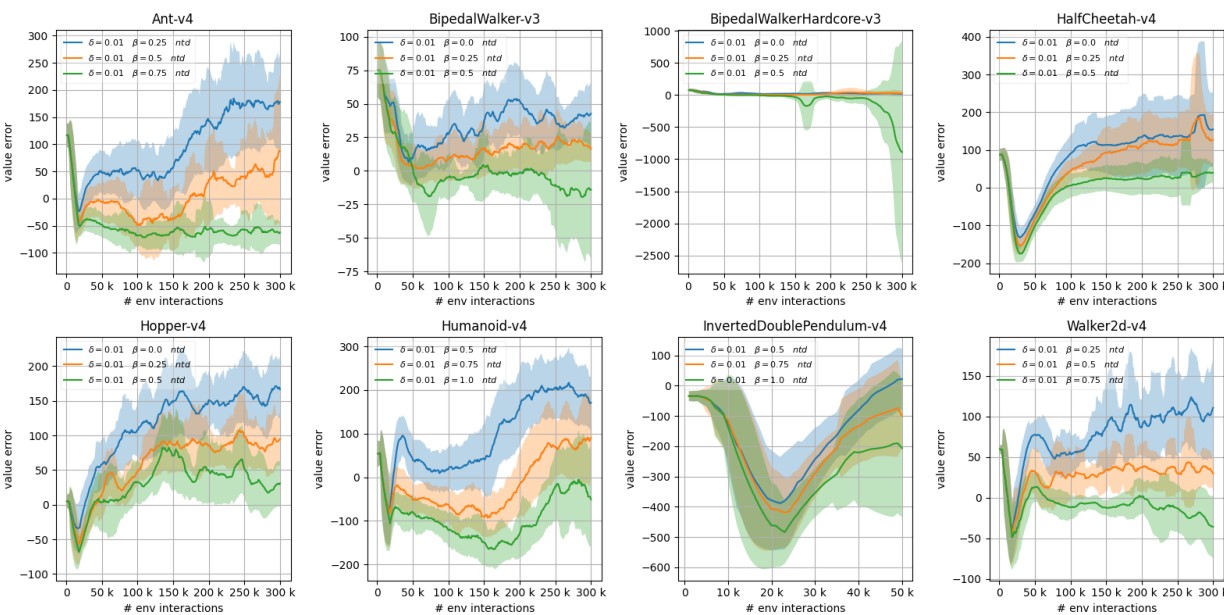

Figure 14: Episodic value estimation error curves of STAC with varying pessimism ($\beta$) parameter. Dropout is equal to 0.01, but target critic dropout is turned off.

## B.2   Dropout Effect

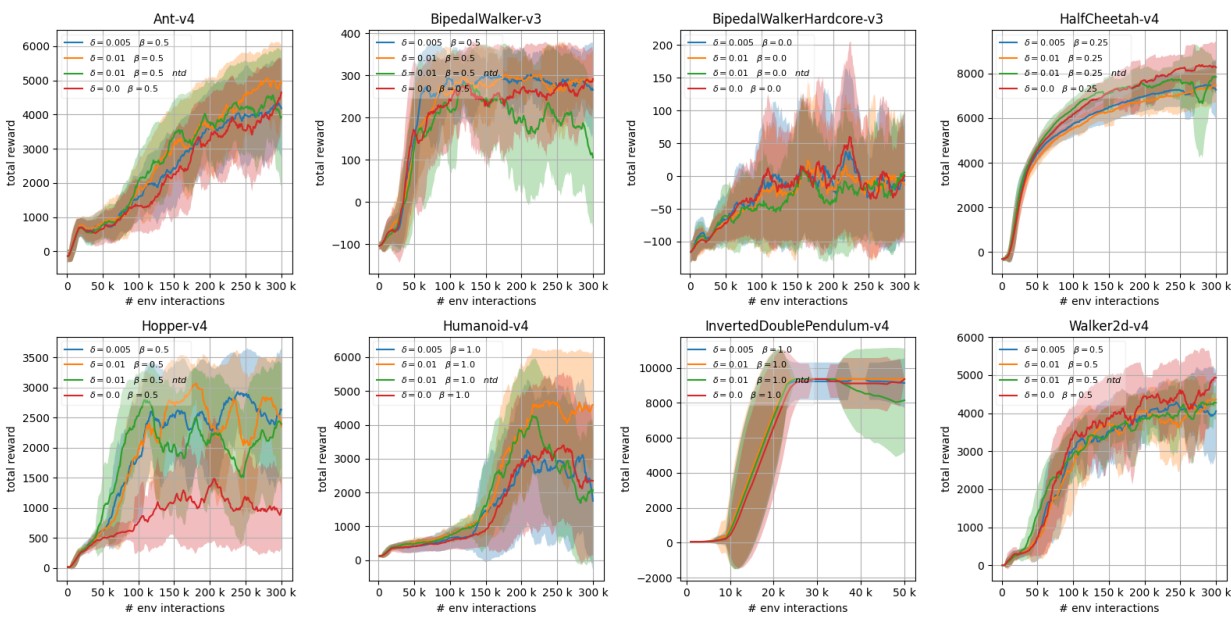

Figure 15: Learning curves of STAC with dropout on and off (for both critic and policy). *ntd* refers to turned off target critic dropout. Pessimism parameters are used the same as the main experiment, available in Table 5.

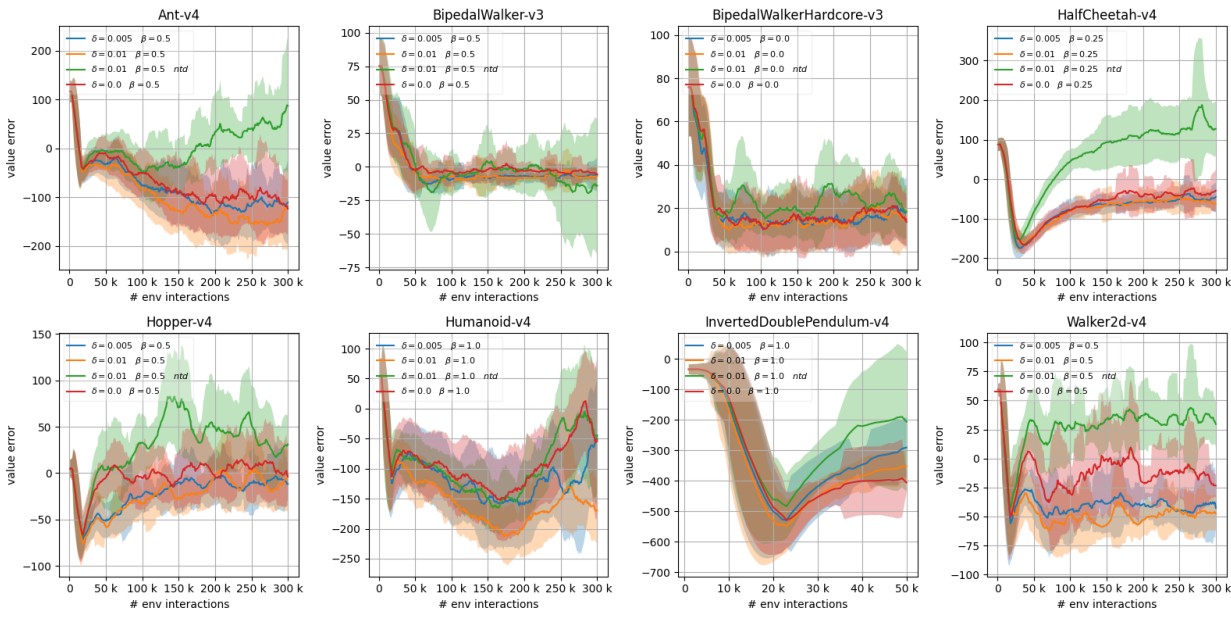

Figure 16: Episodic value estimation error curves of STAC with varying dropout on and off (for both critic and policy). *ntd* refers to turned off target critic dropout. Pessimism parameters are used the same as the main experiment, available in Table 5.

## Appendix C    Hyper-parameters and Experiment Details

Hyper-parameter values used in the experiments per method are listed in Table 4. Dropout parameter is found by trial-and-error and it matches the selection in DROQ paper (Hiraoka et al., 2021). In addition, target entropy and pessimism parameters (only for STAC) are summarized in Table 5. Target entropy values are taken from the DROQ paper, which uses the same values (except `Humanoid-v4`). For STAC, pessimism hyper-parameter and for TQC, quantile drop parameters per environment are found by trial-and-error to obtain the best performance.

Table 4: Experimental Parameters per Algorithm

| Algorithm | Parameter | Value |
|---|---|---|
| STAC, DROQ, SAC, TOPSAC, TQC | Optimizer | Adam ((Kingma & Ba, 2014)) |
| | Critic Learning Rate | $1 \times 10^{-3}$ |
| | Actor Learning Rate | $3 \times 10^{-4}$ |
| | Discount Rate ($\gamma$) | 0.99 |
| | Target-Smoothing Coefficient ($\rho$) | 0.995 |
| | Replay Buffer Size | $1 \times 10^6$ |
| | Mini-Batch Size | 256 |
| | Random Starting Data | 10000 |
| | UTD Ratio ($G$) | 1 |
| DROQ | Dropout Rate | 0.01 |
| TOPSAC, TQC | Number of Quantiles | 25 |
| TQC | Ensemble Size | 5 |
| TOPSAC | Bandit Optimism/Pessimism Arms | [-1, -0.5, 0] |
| | Bandit Learning Rate | 0.1 |
| | Bandit Window Size | 10 |

Table 5: Target policy entropy ($\bar{H}$), pessimism ($\beta$ for STAC), dropout rate (for STAC) and quantile drop ($n_{drop}$ for TQC) per environment, yielding best results

| Environment | Entropy ($\bar{H}$) | Pessimism ($\beta$) | Dropout | Quantile Drop ($n_{drop}$) |
|---|---|---|---|---|
| `Ant-v4` | -4 | 0.5 | 0.01 | 5/25 |
| `BipedalWalker-v3` | -2 | 0.5 | 0.005 | 3/25 |
| `BipedalWalkerHardcore-v3` | -2 | 0.0 | 0.00 | 1/25 |
| `HalfCheetah-v4` | -3 | 0.25 | 0.00 | 0/25 |
| `Hopper-v4` | -1 | 0.5 | 0.005 | 5/25 |
| `Humanoid-v4` | -8 | 1.0 | 0.01 | 12/25 |
| `InvertedDoublePendulum-v4` | -1 | 1.0 | 0.01 | 3/25 |
| `Walker2d-v4` | -3 | 0.5 | 0.00 | 5/25 |

## Appendix D    Network Architectures of STAC

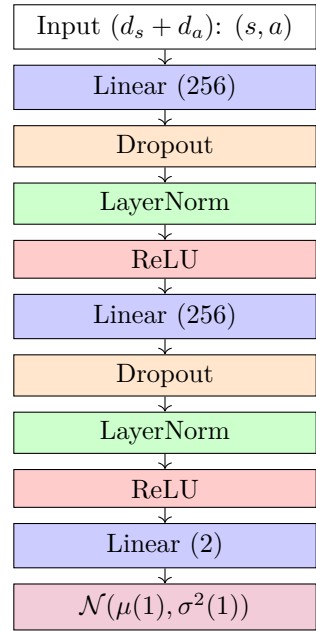

Figure 17: Critic network architecture

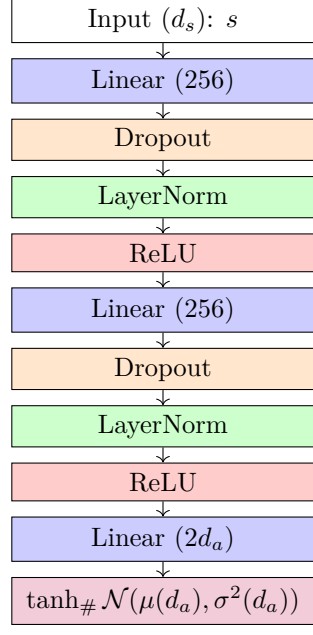

Figure 18: Policy network architecture

## Appendix E    Source Code

Our results can be accessed publicly at `https://github.com/authors-github/stochastic-actor-critic-results`. This code uses our in-house developed RL framework as a sub-repository, available on `https://github.com/authors-github/rl-warehouse`.

