# OpenReview forum: "Simplifying Actor-Critic Reinforcement Learning: Mitigating Overestimation Bias with a Single Distributional Critic"
_TMLR — Rejected by TMLR_

### Review · Reviewer_hmSK · 2025-01-30

**Summary Of Contributions:**

The paper presents a well-motivated approach to mitigating critic overestimation in actor-critic reinforcement learning. STAC leverages a single distributional critic network, Bayesian dropout for regularization, and a carefully tuned pessimism parameter (β) to balance stability and exploration. A key distinction from previous methods is the use of aleatoric uncertainty—rather than epistemic uncertainty—for pessimistic updates. This approach avoids excessive conservatism while maintaining efficiency. The computational advantage of using a single critic instead of ensembles makes STAC a practical alternative to state-of-the-art pessimistic RL algorithms.

**Audience:**

Yes

**Claims And Evidence:**

Yes

**Requested Changes:**

- More discussion on adaptive methods for tuning β would further strengthen the approach.

 - As sample efficiency is a key aspect, comparing STAC to REDQ (a strong baseline for efficient off-policy learning) and other ensemble-based methods like SUNRISE would be insightful. It would help highlight whether STAC can match or surpass them in sample efficiency with fewer networks.

  - In environments with high stochasticity or sparse rewards, could over-pessimism still hinder learning? A discussion about how does STAC perform in such settings compared to ensemble-based methods can be valuable

**Strengths And Weaknesses:**

## **Strengths**
- **Novel Handling of Critic Overestimation**
  STAC uniquely leverages aleatoric uncertainty instead of epistemic uncertainty for pessimistic updates, avoiding the need for ensembles while maintaining stability.

- **Efficient & Robust Distributional Critic**
 The heteroscedastic distributional critic captures aleatoric uncertainty, improving robustness and learning stability.  Unlike ensemble-based methods, STAC maintains high sample efficiency using a single critic.

-  **Balanced Pessimism for Policy Learning**
   The paper argues that full overestimation prevention is unnecessary, tuned pessimism (β) balances exploration and stability.  The empirical results support that β should be environment-specific where less pessimism for easier tasks, more for harder ones.
-  **Dropout Regularization for Exploration & Stability**
 Bayesian dropout is used for both critic and policy networks, promoting regularization and exploration without excessive conservatism.


## **Weaknesses & Areas for Improvement**

- **Hyperparameter Sensitivity and Tuning Complexity**
- While the authors acknowledge the need for careful tuning of pessimism (β) and dropout rate, the lack of a principled way to automatically adapt these parameters could be a limitation in real-world applications.
- More discussion on adaptive tuning methods for β would strengthen the paper.

- **Trade-Off Between Pessimism and Exploration**
- The paper correctly argues that pessimism should not hinder exploration.
- However, it would be beneficial to analyze cases where over-pessimism could still lead to suboptimal exploration in highly stochastic or sparse reward environments.

- **Lack of Theoretical Justification for Dropout Rate Selection**
- The authors demonstrate empirically that different dropout configurations yield similar pessimism effects.
- However, a more theoretical analysis on optimal dropout rate selection would make the approach more robust and generalizable.

- **Comparison to Other Pessimistic RL Methods**
- The paper primarily compares STAC to SAC, DROQ (G=5), TOPSAC, and TQC but does not provide a detailed evaluation against other pessimistic RL methods such as REDQ or Conservative Q-Learning.
- A broader comparison, especially against REDQ, would strengthen the claims of efficiency and effectiveness.

---

> ### Author Response · Authors · 2025-02-05
>
> We appreciate your valuable comments. We tried to explain our approach as answer to some parts of weakness & areas for improvement part.
>
> - **Hyperparameter Sensitivity and Tuning Complexity**: We previously thought about using an adaptive mechanism for β (pessimism). However, we choose to focus on pessimism in the face of aleatoric uncertainty approach, keeping the algortihm simple. In previous methods, TOP algorithm employed a bandit mechanism to update pessimism. However, this kind of methods use on-policy returns to update bandit and this makes the method unusable for offline RL. Although we do not focus on offline RL but off-policy online learning, STAC can be extended to fully offline methods in future works.
>
> - **Trade-Off Between Pessimism and Exploration**: We think that `BipedalWalkerHardcore-v3` environment is kind of sparse reward environment. A very specific maneuver is required to tackle big hurdles and pits on the terrain. It has also stochastic nature since terrain is randomly generated at each episode, and the walker is not aware of what is behind of hurdles. The results are discussed in depth under Section 6.1 in the new revision.
>
> - **Lack of Theoretical Justification for Dropout Rate Selection**: Dropout itself is an empirical method for regularization and epistemic uncertainty. We do not claim anything about its theory but use it as heuristic for exploration and regularization.
>
> - **Comparison to Other Pessimistic RL Methods**: We thought to include it at first, but decided not to include since it requires too much computation resource due to bigger ensemble and update-to-data ratio, and we have scarce computation resources. It would also make graphs ambigious due to much results shown. Therefore, we prefer not to conduct another method for now. Still, we will be starting an experiment for REDQ soon and we may remove SAC in favor of REDQ in the comparison since SAC is a relatively older algorithm.

---

> > ### Author Response · Authors · 2025-02-14
> > **Update about REDQ results**
> >
> > We currently finished REDQ runs with UTD ratio 1. The results are very similar to the SAC. We cannot run REDQ with high UTD ratio due to limited computation resources. We are concerned about adding too much results in a single graph. Do you have a suggestion about it? Maybe, we can remove SAC to add REDQ. At least, we can just mention that REDQ yields similar results with SAC in the text without a result.

---

> > > ### Comment · Reviewer_hmSK · 2025-02-25
> > > **about REDQ results**
> > >
> > > Thank you for your comment and for conducting the REDQ runs. If the results of REDQ with a UTD ratio of 1 are very similar to SAC, it is reasonable to avoid overcrowding the graph. Instead of adding more curves to the same figure, it would be acceptable to mention in the text that REDQ yields similar performance to SAC without including the full results in the graph.
> > > However, if feasible, I would still encourage including REDQ (even with the UTD ratio of 1) in the graph if it does not cause clutter. Showcasing results with a more recent method like REDQ can strengthen the paper by demonstrating that the proposed approach remains competitive more recent methods. Alternatively, if including REDQ in the graph is not practical, a brief discussion in the text, supported by a table or supplementary material, could also work well.

---

> > > > ### Author Response · Authors · 2025-02-25
> > > >
> > > > Thank you for the response. We prefer not to add extra results, so as not to overcrowd the graphs/tables. We have two options,
> > > >
> > > > - In the text, mention that REDQ performs similarly to SAC. (In the new revision, under first paragraph of Section 6: Experiments)
> > > > - Remove SAC results, add REDQ results, and mention that SAC performs similarly to REDQ.
> > > >
> > > > We will stick to the first option for now. We will be uploading a new revision in one or two days.

---

### Review · Reviewer_LfAJ · 2025-01-30

**Summary Of Contributions:**

This paper proposes an approach to reduxing overestimation bias in deep
RL by deriving pessimistic value estimates via the aleatoric uncertainty
in the returns. Particularly, assuming return distributions are
sub-gaussian, the authors can bound the overestimation error, which can
be used to derive corrected critic estimates. The paper proposes an
approach under which Gaussian envelopes of the return distributions are
modeled by deep neural networks, which are trained towards
pessimistically-shifted targets via maximum likelihood. Additionally,
the authors demonstrate that layer normalization and dropout can
positively influence performance in their setting. Ultimately, the
authors derive an actor-critic algorithm using the aforementioned critic
architecture, achieving strong performance in a variety of continuous
control domains.

**Audience:**

Yes

**Broader Impact Concerns:**

No broader impact concerns

**Claims And Evidence:**

Yes

**Requested Changes:**

1.  There is a typo in the title – "Distrubitonal".
2.  You have not cited *any* of the foundational distributional RL
    works. At the very least, "Distributional Reinforcement Learning" by
    Bellemare, Dabney, and Rowland should be cited.
3.  Nit: the notation e.g. $\tau:
         \mathcal{S}\times\mathcal{A}\mapsto\mathcal{P}(\mathcal{S})$
    should instead be
    $\tau: \mathcal{S}\times\mathcal{A}\to\mathcal{P}(\mathcal{S})$
    (same for, e.g., $R:\mathcal{S}\times\mathcal{A}\mapsto\mathbb{R}$).
4.  In section 2.2, you say that the Bellman backup
    $\mathcal{T}^\pi Q(s, a)$ is also called the TD target, but this is
    not true. The TD target is computed from samples of state
    transitions (you don't and cannot compute the expectation in
    equation 4 in general). If you could compute $\mathcal{T}^\pi
         Q(s, a)$, you wouldn't really need TD learning.
5.  Nit: above equation 5, you say that the optimal policy is defined as
    the softmax on the optimal Q-function. Firstly, you have not defined
    what the optimal Q-function is yet. Moreover, it is counterintuitive
    to define the optimal policy this way (while it's not incorrect)—I
    think it would make more sense to define it as the policy that
    induces the pointwise greatest Q-function, and in the maximum
    entropy framework this policy is unique and has a nice closed form
    (the second argument of the KL in equation 5).
6.  Just under equation 5, please clarify what you mean by "available
    critic". I think what you mean is "\[…\] solves Equation 5 for the
    estimated critic $Q$ \[…\]".
7.  In section 3.2, you claim that epistemic uncertainty can be reduced
    by "refining the model, or simply using better modeling techniques".
    Can you give examples of this?
8.  In equations 11 and 12, the "stochastic Bellman backups" consume
    distributions over Q-functions and output Q-functions. In other
    words, the RHS of (11) is an element of
    $\mathbb{R}^{\mathcal{S}\times\mathcal{A}}$, while you're trying to
    model
    $\mathcal{Q}\in\mathcal{P}(\mathbb{R}^{\mathcal{S}\times\mathcal{A}})$.
    So how are equations 11 and 12 used to update $\mathcal{Q}$?
9.  Similarly, there must be a mistake in equation 14—you have defined
    $\mathcal{T}^*\matcal{Q}$ when $\mathcal{Q}$ is a distribution over
    functions, but then it's not clear how $\mathcal{T}^*\mu$ is defined
    since $\mu$ is just a function (and not a distribution over
    functions).
10. Following the last point, I'm not understanding what $\mathcal{Q}$
    is even supposed to represent. Is this encapsulating epistemic
    uncertainty or aleatoric uncertainty? Equation 11, as in my previous
    point, makes this entirely unclear. My best guess is that it's
    supposed to encode aleatoric uncertainty, but I really think this
    should be clarified (maybe that's possible by just making the
    notation/math more precise as suggested in the points above).
11. Just above Theorem 4.1, it says "Finally, we possess Theorem 4.1". I
    really don't think "possess" is the right word here. In particular,
    it's very unclear if this theorem is a contribution of the paper, or
    something that already exists in the literature. Upon reading the
    appendix, I see its proof—I recommend pointing to the proof in the
    main text.
12. In the proof of Theorem 4.1, in your integrals over the Q-function
    posterior, the differential should be $dq$ instead of
    $d\mathcal{Q}$.
13. In the third step of the proof of Theorem 4.1, you need some
    assumptions on $\mathcal{Q}$ in order to apply Tonelli's theorem
    (e.g., you need some integrability / boundedness guarantees).
    Moreover, without assuming that $\mathcal{S},\mathcal{A}$ are finite
    (and I see no such assumption), you should be careful integrating
    over $\mathbb{R}^{\mathcal{S}\times\mathcal{A}}$ (an infinite
    dimensional space).
14. At the beginning of section 5.2, it says "Assuming critic value
    distribution is normal \[…\]". Is it reasonable to assume this
    distribution is Gaussian? What are the consequences of this
    assumption? I am guessing this has to do with the fact that for a
    given variance proxy, the Gaussian is sort of a worst case, but then
    does this assumption induce *too much* pessimism? This should be
    clarified.
15. In the neural network diagrams of Appendix D, the critic network
    appears to output only a scalar, but from my understanding it should
    be producing two scalars ($\mu_\theta(s, a)$ and
    $\sigma_\theta(s, a)$).
16. The supplementary material does not appear to actually contain the
    implementation of STAC (or the experimental procedure). It looks
    like you forgot to include a `scripts` directory.
17. Why are the MuJoCo agents trained for only 300k steps? Isn't it
    standard to report performance after 1M training steps?
18. Sections 6.1 and 6.2 defer their experimental results to the
    appendix—this is not very nice, because these sections are clearly
    incomplete without these experimental results.

**Strengths And Weaknesses:**

## Strengths

I found the motivation for this work to be quite interesting, and to the
best of my knowledge, it is novel. In particular, the authors claim that
non-realizability can manifest itself as aleatoric uncertainty, and
pessimism with respect to this source of uncertainty could be crucial
for value estimation in deep RL. I appreciate this insight, and as far
as I'm concerned, it has not yet been investigated. The hypothesis is
plausible, in my opinion, and I was curious to know if accounting for
aleatoric uncertainty for this reason positively influences deep RL
methods.

## Weaknesses

There are two major concerns in this work, as I see it. I will describe
each below, and more precise details can be found in the requested
changes.

### Clarity of Proposed Approach

In several areas throughout the text (see Requested Changes), some of
the math is not described with enough precision, leaving my with
reasonably high uncertainty about what the proposed method is actually
even modeling. Many of the equations of, e.g. loss functions for STAC,
do not clearly correspond to the (distributional) Bellman operators that
preceed them. Likewise, the neural network architecture diagrams given
in the appendix to not appear to actually represent their function as
described in the main text exactly. To make matters worse, the authors
appear to have left out the source code for STAC in the supplementary
material (it looks like a whole directory is missing), so I could not
check their implementation to resolve any difficulties I had in
understanding the method.

Moreover, some of the theoretical motivation is not explained in enough
detail, even though I largely found it intriguing. For instance,

1.  Why focus on maximum entropy RL? Entropy regularization appears to
    be orthogonal to the issues you discuss in the paper.
2.  Why do you need to model epistemic uncertainty at all? I understand
    that you want to employ pessimism in the face of aleatoric
    uncertainty, but techniques from distributional RL are already
    attempting to only model aleatoric uncertainty. Since you don't
    appear to be using epistemic uncertainty estimates for anything, it
    is not clear to me why it is an important inclusion in the paper and
    the method (besides possibly generic regularization arguments for
    deep neural nets).
3.  While I understand the heuristic to be pessimistic with uncertainty
    that arises due to non-realizability of the value function, the
    heuristic sounds less convincing when you consider highly noisy
    MDPs. If returns inherently have lots of aleatoric uncertainty, do
    you still want to be pessimistic with respect to it?

### Experimental Results

There are a number of concerns I have with the experimental results:

1.  The authors use in-house implementations of the baselines "to make a
    fair comparison", but as noted above, their implementation is not
    provided.
2.  It appears that the agents are not trained for a standard duration:
    300k steps for MuJoCo, last I checked, is not very much—I was
    expecting something more like 1M steps.
3.  Table 1 bolds "significant scores", but it's not clear what
    "significant" means in this context. Also, the table does not report
    confidence intervals (nor are they given in the main text).
4.  With regard to confidence intervals, checking the appendix, we see
    that they are quite large and overlapping. This makes it basically
    impossible to distinguish which method is best.
5.  Likewise, most of the experimental results are relegated to the
    appendix. This is not appropriate, because it is not actually
    possible to interpret the results of the paper without reading the
    appendix. This should have either been a long submission, or the
    organization of the main text could have been changed to accommodate
    the empirical results.
6.  With regard to the dropout results, it appears impossible to
    identify anything statistically significant. As I suggested in the
    "Clarity of Proposed Approach" section, it is not even clear to me
    why dropout *should* be impactful in this setting (aside from
    generic regularization arguments), so these results do not resolve
    my confusion about why so much of the text focuses on it.

---

> ### Author Response · Authors · 2025-02-05
>
> We really appreciate your extensive comments and hope to fix missing/ambiguous parts of the paper. First, sorry about the missing code. We thought that two zip files are uploaded but it seems that one of them failed. Now, we uploaded the codebase and results by merging them into a single zip file.
>
> ## Weaknesses and Questions
>
> Let us answer some of your questions and concerns.
>
> Q1) Why focus on maximum entropy RL? Entropy regularization appears to be orthogonal to the issues you discuss in the paper.
>
> A1) There is no specific reason for this. REDQ paper (Chen et. al.) analyzed the post-update estimation bias in critic ensemble case, which includes `max` operator upon Q distribution (in ensemble form). Similar to them, we prefered to use `logsumexp` operator on Q distribution, which is the natural result of maximum entropy policy improvement. `logsumexp` upon critic distribution appears as `log` of moment generating function of it. Then assuming critic distribution as sub-Gaussian yield the post-update estimation bias bound.
>
> Q2) Why do you need to model epistemic uncertainty at all? I understand that you want to employ pessimism in the face of aleatoric uncertainty, but techniques from distributional RL are already attempting to only model aleatoric uncertainty. Since you don't appear to be using epistemic uncertainty estimates for anything, it is not clear to me why it is an important inclusion in the paper and the method (besides possibly generic regularization arguments for deep neural nets).
>
> A2) As you stated, dropout is not used other than regularization at the first place. However, our argument is epistemic uncertainty can be used for `exploration`, which is the main idea of optimism in the face of the uncertainty principle. Most of actor-critic methods use epistemic uncertainty for pessimism to mitigate critic overestimation, revealing the necessity of a high update-to-data (UTD) ratio. This is also the main reason for difficulty in devising an optimistic actor-critic algorithm in our opinion. We believe that epistemic uncertainty should be used for optimism (so for exploration) like bandits. In short, the main idea of STAC `pessimism in the face of aleatoric uncertainty` and `optimism in the face of epistemic uncertainty`. We tried to make this distinction but did not leave a section for this since we used dropout only in a heuristic way. However, this detail will be explained in the Section 1.2 in the new revision.
>
> Q3) While I understand the heuristic to be pessimistic with uncertainty that arises due to non-realizability of the value function, the heuristic sounds less convincing when you consider highly noisy MDPs. If returns inherently have lots of aleatoric uncertainty, do you still want to be pessimistic with respect to it?
>
> A3) In MuJoCo environments, difficult environments require more pessimism (`Humanoid-v4`, `Ant-v4`) while easier ones need less (`HalfCheetah-v4`). The difficulty can be assessed by the nonlinearity of environemt/reward. However, comparing `BipedalWalkerHardcore-v3` (difficult) to  `BipedalWalker-v3` (easy), the difficult one has better performance with zero pessimism. It signals that overpessimism on highly stochastic environments hinders exploration too, as you suspected. However, recall that `HalfCheetah-v4` also has the best performance with zero pessimism although it is a deterministic and easier task. If there is a very slight (or no) overestimation phenomenon, most of the aleatoric uncertainty would be sourced by environment/reward stochasticity. In this case, we believe that the agent should act `neutral` (since the aim is to maximize average return, not to maximize maximum/minimum possible return). Therefore, aside from stochasticity, it is ok to be neutral but not optimistic upon aleatoric uncertainty. We created a paragraph for this under Section 6.1.

---

> > ### Author Response · Authors · 2025-02-05
> >
> > cont'd
> >
> > Q4) The authors use in-house implementations of the baselines "to make a fair comparison", but as noted above, their implementation is not provided.
> >
> > A4) Implementation is uploaded.
> >
> > Q5) It appears that the agents are not trained for a standard duration: 300k steps for MuJoCo, last I checked, is not very much—I was expecting something more like 1M steps.
> >
> > A5) REDQ (Chen et. al.) and MBPO (Janner et. al.) papers used 300k steps as they converge faster compared to low UTD methods. To be honest, we have scarce computation resources and used 300k steps since we observed that STAC nearly converged in all environments with that much duration.
> >
> > Q6) Table 1 bolds "significant scores", but it's not clear what "significant" means in this context. Also, the table does not report confidence intervals (nor are they given in the main text).
> >
> > A6) We did not want to assess a single best method by slight differences in the score but determined the best methods. We will be explaining this in the related part of the text. We will also be adding mean+std scores at the end of learning(300kth step).
> >
> > Q7) With regard to confidence intervals, checking the appendix, we see that they are quite large and overlapping. This makes it basically impossible to distinguish which method is best.
> >
> > A7) We will increase the smoothing parameter of all reward graphs.
> >
> > Q8) Likewise, most of the experimental results are relegated to the appendix. This is not appropriate, because it is not actually possible to interpret the results of the paper without reading the appendix. This should have either been a long submission, or the organization of the main text could have been changed to accommodate the empirical results.
> >
> > A8) Main comparison results will be taken into the text.
> >
> > Q9) With regard to the dropout results, it appears impossible to identify anything statistically significant. As I suggested in the "Clarity of Proposed Approach" section, it is not even clear to me why dropout should be impactful in this setting (aside from generic regularization arguments), so these results do not resolve my confusion about why so much of the text focuses on it.
> >
> > A9) We tried to answer it in A2. In Figure 12, the dropout rate has a significant impact for `HalfCheetah-v4`, `Hopper-v4` and `Humanoid-v4`.

---

> > > ### Author Response · Authors · 2025-02-05
> > >
> > > ## Requested Changes
> > >
> > > We will be fixing typos and notation-related things based on your response. However, we want to inform you about some of the requests.
> > >
> > > R7) We think that a neural network with more generalization would have low epistemic uncertainty even in out-of-distribution samples since it correctly estimates these samples. In new revision, we mentioned this shortly: This type of uncertainty can be reduced by gathering more data or using a better model with higher generalization capability.
> > >
> > > R8) It is true, there is an ambiguity about it. For this, we defined expected Bellman backup instead of stocastic Bellman update, i.e., $\mathcal{T}^{\*} \mathcal{Q}(s,a)$  -> $\mathbb{E}_{Q \sim \mathcal{Q} } [ \mathcal{T}^{\*} Q(s,a) ]$.
> > >
> > > R10) It represents predictive uncertainty (total uncertainty) since any type uncertainty of critic may be exploited when the Bellman optimality operator $\tilde{T}^{\*} Q$ is employed. We noted it on end of first paragraph of section 4. However, our decision is to use only aleatoric part of it for the sake of pessimism since we assume that most of predictive (total) uncertainty is consisted of aleatoric uncertainty. We believe that using pessimism upon epistemic uncertainty would hinder exploration, as discussed in A2.
> > >
> > > R12) It is true, $\mathcal{Q}$ is a probability density function, so it cannot be used instead of probability measure in the expectation. In new revision, we used expectation form to avoid confusion.
> > >
> > > R13) In new revision, we assumed that $\mathcal{Q}(s,a)$ distribution has a finite support for all state-action pairs, in the Theorem 4.1. Therefore, the integrand has an finite upper bound.
> > >
> > > R14) For implementation, we decided to use Gaussian distribution. This converts second inequality of Proof of Theorem 4.1 into an equality. It tightens the bound and increases overestimation, i.e., it covers the worst case as you stated. We mentioned this in the text in new revision.
> > >
> > > R15) In the diagram, we used 1 to emphasise the network returns a distribution over 1-dimensional variable. Same for policy, a distribution over $d_a$-dimensional variable (action space). In the new revision, we adjusted output as follows:
> > >
> > > - Q distribution network: $\mathcal{N} (\mu(1), \sigma^2(1))$
> > > - Policy network: $ \tanh_{\\#}\mathcal{N} ( \mu(d_a), \sigma^2(d_a) )$

---

> > > > ### Comment · Reviewer_LfAJ · 2025-02-18
> > > >
> > > > Thanks to the authors for their detailed responses and their amendments to the draft.
> > > >
> > > > 1. **Epistemic uncertainty and Dropout**: Thanks for clearing this up. I still suspect to much of the text is focused on this part (this is what led to my confusion), especially since it is, as you say in your response, just a heuristic. Moreover, I am not convinced that the experimental results show that this is statistically helpful. One suggestion---based on my limited exposure to Thompson-sampling approaches to RL, I would expect you to effectively sample a Q function (or in your case, a return distribution function) at the beginning of the episode, and act according to that sample for the entirety of the episode (as opposed to randomly sampling a Q function at each step). I believe this is what leads to deep exploration (see e.g. the anthology of https://arxiv.org/abs/1703.07608).
> > > > 1. **Dealing with inherent stochasticity in the environment**: Your comments on this are effectively reinforcing my point: in general, you cannot just act pessimistically with respect to your aleatoric uncertainty estimate. This leads me to believe that the claims of the paper hold really only for deterministic MDPs (and deterministic policies?). This is fine, since most RL benchmarks are described by deterministic MDPs, but this should be clarified.
> > > > 1. **Overlapping confidence intervals**: Smoothing the curves is not an appropriate solution here, that will just hide/obfuscate the data. The issue is that your confidence intervals suggest that there is not a statistically significant difference in performance (in most environments) between the agents you're testing.
> > > > 1. **R8**: I still do not understand what this operator is doing. What is the use of an operator returning a scalar-valued function when you're trying to estimate a distribution-valued function?
> > > > 1. **R13**: Do you mean _bounded_ support? Finite support would mean that the distribution is supported on a finite set, which seems unlikely.

---

> > > > > ### Author Response · Authors · 2025-02-23
> > > > >
> > > > > We thank to the reviewer for interest and improvements on the paper. Here are the detailed explanations for the questions and concerns.
> > > > >
> > > > > A1) Actually, we aimed to focus on overestimation mitigation by pessimism on aleatoric uncertainty. Other than exploration, we focused on epistemic uncertainty to demonstrate the difference from aleatoric uncertainty, and previous works usually used epistemic uncertainty for overestimation mitigation. Since we observed using dropout improves performance on some environments, we state that `dropout enhances exploration` according to previous works (Gal et al. https://www.cs.ox.ac.uk/people/yarin.gal/website/PDFs/DeepPILCO.pdf), but we do not prove this either theoretically or empirically. It is just possible that dropout allows a better regularized learning procedure and improves performance, rather than exploration. We indicated that dropout is just a regularization/exploration heuristic in the next revision. We did also shorten the `Epistemic Uncertainty` section more, removed variatonal inference part.
> > > > >
> > > > > In addition, work of Osband et al. (https://arxiv.org/abs/1703.07608) focuses on discrete RL, while the `Q` function is also policy. In STAC, the policy network has also dropout, and actions taken with dropout include epistemic uncertainty of policy network. This part can be converted to episodic exploration as you stated, i.e., policy dropout mask can be fixed throughout the episode. However, we prefer not to dive in this topic as the main topic is overestimation mitigation using aleatoric uncertainty, dropout is used as a regularization/exploration heuristic.
> > > > >
> > > > > A2) This is open to research for future works. In STAC, critic overestimation and environment/reward stochasticity blend in a single Normal distribution. Therefore, pessimism should be less (close to zero, but still not positive) in highly stochastic environments. However, we still believe that it is useful for overestimation mitigation (`BipedalWalker-v3` is also stochastic environment but still requires significant amount of pessimism, and `BipedalWalkerWalker-v3` also tends to learn with $\beta=0.25$ in more stable way but slower than $\beta=0.0$, see Figure 11). As we stated in the future work part, pessimism due to aleatoric uncertainty can be balanced by optimistic exploration upon epistemic uncertainty. In the next revision, we will indicate that our claim for stochastic environments still requires detailed experiments, in `Conclusion` section.
> > > > >
> > > > > A3) The best method other than STAC is TQC according to our experiments, and there is no significant performance difference between them in general, as you state. The good performance of TQC comes from the same idea, a number of critic quantiles to drop during learning as the pessimism parameter specifically selected for the environments. Rather than score performance (sample efficiency), we think that STAC is better than TQC since it performs similar in score, while it uses a single critic network and a Normal distribution as critic output, and it consumes less computation power and memory.
> > > > >
> > > > > A4) As you state, $\mathcal{T}$ is used on deterministic `Q` functions. On policy improvement step, a scalar `Q` value is sampled, and this `scalar sample` is used to improve policy. In standart actor-critic learning, the resulting action is $\arg\max_{a} Q(s, a)$ while resulting value is $\max_{a} Q(s, a)$ on state $s$. If many sampled `Q` values are used to improve policy, the resulting value would be
> > > > > $\mathbb{E}_{Q \sim \mathcal{Q}} [\max_a Q(s, a)] $, after policy update.
> > > > >
> > > > > However, in the ideal case, the resulting value should be $\max_a \mathbb{E}_{Q \sim \mathcal{Q}} [Q(s, a)]$, since the average estimate should be improved. However, this is not possible since each error of sampled `Q` functions are exploited cumulatively and ground truth `Q` function cannot be evaluated due to ongoing policy updates. The arising difference between ideal and estimated Bellman backups allows us to evaluate overestimation error. A similar analysis is conducted by Chen et al. (https://arxiv.org/pdf/2101.05982), for REDQ algorithm.
> > > > >
> > > > > In soft actor-critic form, $\max_a Q(s, a)$ term is replaced by $\alpha \text{lse}_{a} \( {\alpha}^{-1} Q(s, a) \)$, where $\text{lse}$ is the log-sum-exp function. Therefore, after the update, the Bellman backup becomes
> > > > >
> > > > > $\mathcal{T}^*Q(s, a) = r(s,a) + \gamma \mathbb{E}_{s'\sim\tau(\cdot|s,a)} \[ \alpha \text{lse} _{a'} \( {\alpha}^{-1} Q(s', a') \) \] $, in deterministic form.
> > > > >
> > > > > The distributional form is already explained in the paper.
> > > > >
> > > > > A5) Thank you for the correction. We meant the samples from Q distribution has finite values. This allows us to use Tonelli's theorem. In the new revision, this will be updated as `bounded support`.

---

> > > > > > ### Comment · Reviewer_LfAJ · 2025-02-24
> > > > > >
> > > > > > I appreciate the response, though I have a few more questions/comments about these points.
> > > > > >
> > > > > > 1. **Dropout**. Thanks for reducing the epistemic uncertainty section, I think this is the appropriate mitigation, though I still think section 3.2 can be removed entirely (you can comment on the difference between epistemic uncertainty and aleatoric uncertainty in section 3.1).
> > > > > > 1. **Bellman Operator**. I understand that you only use a scalar to update the actor, but I don't understand how you're training the critic. You say "the distributional form is already explained in the paper", but this is the part I'm struggling with. I think I do understand it a little better in the revision, but I am still a little confused. Why are you doing maximum likelihood on the pessimistic Q value (the one where you've already subtracted the standard deviation)? Shouldn't you be estimating the maximum likelihood of the pure TD target? The minimizer of equation (20) will have mean $\mu - \beta\sigma$, but then you train maximum likelihood again on samples with mean $\mu - \beta\sigma$ so the next iteration should have mean roughly $\mu - 2\beta\sigma$, and so on.
> > > > > > 1. **Bounded supports**. It doesn't make sense to simultaneously assume that $\mathcal{Q}(s, a)$ is Gaussian *and* that its support is bounded. I don't think this should be a dealbreaker, but the proof you have given should be formalized.

---

> ### Author Response · Authors · 2025-02-25
>
> Here are the explanations/updates about the latest questions.
>
> - In the new revision, we removed the equations and left two paragraphs for epistemic uncertainty, explaining verbally. We also updated the visual for demonstrating aleatoric and epistemic uncertainty.
>
> - Yes, this is a tricky point. In Theorem 4.1, we analyzed the overestimation of expected Bellman update $\mathbb{E}_{Q\sim \mathcal{Q} } [ \mathcal{T}^{*} Q(s,a) ]$. In the analysis, we did not define policy, as the policy is just softmax of critic values over all actions. Although this is only possible for discrete action space, it is extended to continuous action space (using actor-critic architecture) by learning the softmax of critic over actions using the policy update rule (Equation 5).
>
> Again, in soft Q-learning setting, there is no explicit policy network, it is only softmax over critic action values. Then, we should use pessimistic TD targets to train critic network, according to Corollary 4.1. Coming to the actor-critic case, why do we use pessimistic value in Equation 20 as a policy learning objective then? Because critic training objective and policy improvement objective should be the same. In other words, at learning step $k$, Bellman backup must be equal to Bellman policy evaluation with the new policy, $\mathcal{T}^* Q^k(s,a) = \mathcal{T}^{\pi^{k+1}} Q^k(s,a)$. It is only possible by using pessimistic critic value for policy improvement.
>
> - We think we can assume a sub-Gaussian distribution as bounded. We also assumed critic distribution as sub-Gaussian. However, we modeled the critic network as Gaussian distribution for practical implementation. We clarified this in section 5.1, in the new revision.
>
> We mentioned the latest changes here, but we prefer to wait to upload the new revision for one or two days more, for potential additional feedback from you and other reviewers.

---

> > ### Comment · Reviewer_LfAJ · 2025-02-26
> >
> > Re: boundedness of distributions, thanks for the clarification, this makes sense.
> >
> > I think I understand what you're saying about the critic now. But I maintain my position that the section describing the algorithm can be clarified substantially for the reasons I mentioned previously.

---

> > > ### Author Response · Authors · 2025-02-26
> > >
> > > We also thank you for detailed and quick responses. For the case about using pessimism for both critic and policy updates, we dedicated a paragraph on section 5.2, using explanations similar to our last comment.

---

### Review · Reviewer_ARzF · 2025-02-03

**Summary Of Contributions:**

This paper introduces Stochastic Actor-Critic (STAC), a new reinforcement learning algorithm that can use a single distributional critic to mitigate overestimation bias. Unlike prior approaches that rely on epistemic uncertainty and ensembles, STAC leverages aleatoric uncertainty for pessimistic learning, ensuring stable and sample-efficient training. It also incorporates Bayesian dropout for epistemic uncertainty estimation, improving exploration and regularization. By eliminating the need for multiple critics and maintaining a low update-to-data (UTD) ratio, STAC achieves strong performance on continuous control benchmarks while being computationally more efficient than ensemble-based methods.

**Audience:**

Yes

**Claims And Evidence:**

No

**Requested Changes:**

Please go through the paper and fix the minor issues. Right now, it is hard to read. For example:
- Distrubitonal -> Distributional
- computation efficient -> computation-efficient
- "Prior Art" I'm not sure what this means.
- ablations studies -> ablation studies

Can be good to emphasize the core insight behind your design in earlier part of the paper, ideally with some illustration to quickly convey the main idea to the reader.

Right now the proposed method seems to be a combination of DroQ and TQC, I am a bit concerned about the significance of the contribution, can you conduct other in-depth investigations to make the paper more interesting? At one point authors discussed pessimism can hinder exploration, is there experiments you can do to dive deeper into this claim?

**Strengths And Weaknesses:**

Strengths:
- Some theoretical results are presented.
- Discussion of the experiments is quite clear.
- Hyperparameter settings are provided.
- Table 1 shows STAC has in general same or better performance compared to other compared algorithms.
- A larger number of strong baselines are compared with
- Ablation studies are provided


Weaknesses:
- Too many typo and other small issues in the current version.
- It took too long for me to start to understand the core idea of the paper, it seems this paragraph in section 5 is quite important. " the predictive uncertainty of in-distribution samples comes from aleatoric uncertainty, while out-of-distribution samples are detected by high epistemic uncertainty. Out-of-distribution samples are important and should not be disregarded by pessimism. On the contrary, such samples should be explored using suitable methods." perhaps it's good to discuss the core ideas in earlier part of the intro, and maybe use a figure to illustrate it.
- Good to have a computation efficiency table (with wall-clock time) to show how your method compares to others, I see that table has backdrop passes, but what about forward passes?
- It seems to me it can make sense to say pessimism in previous methods might hinder exploration, but how do we know this is actually a problem?
- Results seem a little weak, Figure 3 it seems it shows STAC can have large negative errors in some environments, why is this a good thing? Additionally, although table 1 shows STAC has generally higher performance, its performance is quite similar to TQC in this table and in the performance curves figure. TQC also has a similar distributional design, what is the advantage of TQC?
- Some parts of the paper is unclear, for example, Table 1, how is "Significant scores" defined?
- Table 1, DroQ G=5 I'm pretty sure in the original paper they got much higher performance, I understand the UTD is different but is it really correct that G=5 setting barely improves over G=1?
- I get it that the proposed method can work with a single network and UTD of 1, but the initial idea of having a high UTD is to improve sample efficiency, and to allow high UTD, bias reduction is needed. How does STAC perform under high UTD?

---

> ### Author Response · Authors · 2025-02-05
>
> We thank you for these important comments. Based on your concerns, we updated the paper;
>
> - We fixed the mentioned typos.
>
> - Detailed explanation of usage of epistemic uncertainty with dropout for exploration in depth, with related references in Section 1.2 and 1.1. Additionally, we added a visual to demonstrate the difference between aleatoric and epistemic uncertainty.
>
> - In addition to backward pass, we summarized the number of forward critic calls to build temporal difference target, assuming this is what you meant.
>
> - Previous methods can achieve high performance only by using a very high update-to-data ratio. Probably, lack of exploration is balanced by high UTD ratio, i.e., exploration is realized in loss minimization. With UTD ratio equal to 1, STAC and TQC clearly explores the environment better.
>
> - Negative errors are assumed good since it proves that the value estimate is lower than the episodic return, i.e., there is no overestimation.
>
> - The advantage of TQC is to use ensemble method for epistemic uncertainty and quantile representation for aleatoric uncertainty. STAC uses Bayesian dropout for epistemic uncertainty and heteroscedastic network for aleatoric uncertainty. Ensemble methods are best in terms of performance to our knowledge, but requires more computation resources. This is the advantage and disadvantage of TQC. The only theoretical difference is that TQC uses total (epistemic+aleatotic) uncertainty to mitigate overestimation, while STAC only uses aleatoric uncertainty for this purpose.
>
> - About DROQ, we shared the codebase. We will be investigating if there is an implementation error. However, we could not find a problem for now.
>
> - It is worth investigating how STAC performs under high UTD ratio. We prefer to leave this as future work, as it requires to conduct ablations studies again to obtain the best pessimism and dropout rate.
>
> - In the new revision, we discussed of pessimism effect on highly noisy environments, using results from `BipedalWalkerHardcore-v3` by explaining source of environment uncertainty and reward sparsity.

---

> > ### Comment · Reviewer_ARzF · 2025-03-03
> > **Thank you for your response**
> >
> > Thank you for the rebuttal, I can see the paper has been improved a lot.
> >
> > Some other comments:
> > - "Negative errors are assumed good since it proves that the value estimate is lower than the episodic return, i.e., there is no overestimation." I get that it shows no overestimation, but why would underestimation be good compared to, say, accurate estimation?
> > - "With UTD ratio equal to 1, STAC and TQC clearly explores the environment better." How do you know they clearly explore the environment better?

---

> ### Author Response · Authors · 2025-03-03
>
> We also thank you for assessing our paper and previous answers. Our explanations to your latest comments:
>
> - The best case is surely accurate value estimation. Where STAC yields negative errors (underestimate), the other methods overestimate the value function. Underestimation slows learning, but overestimation causes getting stuck in a local minimum, which is not recoverable.
>
> - STAC and TQC perform better with UTD ratio 1 compared to other methods with the same UTD ratio. Maybe this is the right sentence to say. The reason for the better performance is accurate value estimation. Accurate value estimation guides the policy towards the right states to explore. With a worse value estimation, learning stucks/saturates and novel states cannot be explored anymore.

---

### Decision · Action_Editor_otyA · 2025-03-19

**Recommendation:** Reject

**Comment:**

The paper explores an interesting and relevant topic in reinforcement learning. The approach toward handling these issues of overestimation bias (and over-conservatism) is also novel.

There are several factors that need to be improved before the paper is ready for publication.

There are some problems with the motivation and clarity as pointed out by one reviewer in depth in their review and responses (and recommendation). Another reviewer also mentioned that the claim of negative bias is beneficial and can help exploration but the only way this is substantiated is via better performance.

The results are problematic for several reasons:

- First, as eluded to in the previous point, the paper makes several arguments about the importance of aleatoric and epistemic uncertainty, but then provides evidence only in the form of improved performance (and only one one domain). The improved performance could be due to other reasons, such as a dropout heuristic, or use of distributional RL. Experiments that demonstrate the stated importance of the uncertainty sources more directly would provide stronger evidence, and this could be done on another domain (that is perhaps easier than Mujoco) especially chosen for demonstrative purposes.

- Second, there is a repeated claim that the number of steps does not line up with what is standard for this domain (300k versus 1M). The authors argue that most of the graphs have converged by that point and previous work used that value. This would again not be as significant a concern if there were more domains (as suggested above) to help support the claims more clearly. With just the one domain and performance as the only justification for the claims, it is a fair criticism to expect that the results be run to the standard number of steps, especially when several of them have clearly not converged.

- Third, as stated in one review: many of the results required to understand the paper are in the appendix. The appendix should be designated for extra/supplementary results only; the main results to substantiate the claims should be in the main paper.

- Fourth, there are issues with statistical significance which makes the performance-only argument used to support the claims even weaker.

Presentation is a major outstanding issue. One recommendation stated that it needs more careful writing. I agree with one of the reviewer's comments "It took too long for me to start to understand the core idea of the paper". I tried to read the paper and really count not understand what the main idea was after Section 1. I will admit that it is partly because the paper takes for granted that the reader is well-familiar with the terms "aleatoric uncertainty" and "epistemic uncertainty".. in a paper where these concepts are central to the paper, there should be one-sentence definitions of them very early in the paper (e.g. when looking this up, I found this paper which describes both in the first three sentences of the abstract: https://arxiv.org/abs/2206.01558). Presentation is a recurring problem with the paper that caused a fair bit of confusion among reviewers. I believe the author has clarified some of the points through responses and improvements to the paper. I strongly encourage a rewrite of the introduction that sticks to the high-level main idea describe in a somewhat easy-to-understand way (including the necessary concepts to make such a point) before ramping up on the technical discussion of related work. For example, in one response the author stated (with emphasis) "pessimism in the face of aleatoric uncertainty and optimism in the face of epistemic uncertainty". This is a succinct and easy way to understand describe the overall idea. Paired with simple descriptions of those two concepts preceding this sentiment, the paper could go on to explain how this will be achieved and why it's important.

Overall, the paper still needs significant improvement before it's ready for publication, but I do believe this topic is highly relevant to TMLR. Hence, I encourage the author to resubmit a paper with the problems identified above addressed.

**Audience:**

Yes, this is a good fit for the TMLR audience.

**Claims And Evidence:**

After a thorough read of the reviews, responses, and recommendations: no, I do not believe the claims are well-supported.

One of the main criticisms (consistent across recommendations) was the that results provide only weak evidence of the claims.

More detail in the explanation for the recommendation below.

**Resubmission Of Major Revision:**

The authors may consider submitting a major revision at a later time.